# Topographical change caused by moderate and small floods in a gravel bed ephemeral river - a depth-averaged morphodynamic simulation approach

Eliisa Lotsari[1, 5], Mikel Calle[2], Gerardo Benito[2], Antero Kukko[3, 4], Harri Kaartinen[3], Juha Hyyppä[3], Hannu Hyyppä[4] and Petteri Alho[3, 4, 5]

[1]Department of Geographical and Historical Studies, University of Eastern Finland, Joensuu, Yliopistokatu 2, P.O. Box 111, 80101 Joensuu, Finland

[2] National Museum of Natural Sciences, Spanish Research Council (CSIC), Madrid, Calle de Serrano 117, 28006 Madrid, Spain

[3] Department of Remote Sensing and Photogrammetry, Finnish Geospatial Research Institute, National Land Survey of Finland, Kirkkonummi, Geodeetinrinne 2, 02430 Masala, Finland

[4] Department of Built Environment, Aalto University, Espoo, Vaisalantie 8, P.O.Box 15800, 00076 Aalto, Finland

[5] Department of Geography and Geology, University of Turku, Turku, 20014 Turun yliopisto, Finland

*Correspondence to*: Eliisa S. Lotsari (eliisa.lotsari@uef.fi)

**Abstract.** In ephemeral rivers, channel morphology represents a snapshot at the end of a succession of geomorphic changes caused by floods. In most cases, the channel shape and bedform migration during different phases of a flood hydrograph cannot be identified from field evidence. This paper analyzes the timing of river bed erosion and deposition of a gravel bed ephemeral river channel (Rambla de la Viuda, Spain) during consecutive, moderate- (March 2013) and low-magnitude (May 2013) discharge events, by applying a morphodynamic model (Deflt 3D) calibrated with pre- and post-event surveys by RTK-GPS points and mobile laser scanning. The study reach is mainly depositional and all bedload sediment supplied from adjacent upstream areas is trapped in the study segment forming gravel lobes. Therefore, estimates of total bedload sediment mass balance can be obtained from pre- and -post field survey for each flood event. The spatially varying grain size data and transport equations were the most important factors for model calibration, in addition to flow discharge. The channel acted as braided channel during the lower flows of the two discharge events, but when bars were submerged in high discharges of May 2013, the high fluid forces followed a meandering river planform. The model results showed that erosion and deposition were in total greater during the long-lasting receding phase than during the rising phase of the flood hydrographs. In the case of moderate-magnitude discharge event, deposition and erosion peaks were predicted to occur at the beginning of the hydrograph, whereas deposition dominated throughout the event. On the contrary, the low-magnitude discharge event only experienced the peak of channel changes after the discharge peak. Thus, both type of discharge events highlight the importance of receding phase for this type of gravel bed ephemeral river channel.

## 1 Introduction

The hydrology of ephemeral rivers is dominated by occasional large flash floods that cause morphological change (Tooth, 2000; Benito et al., 2011). Due to their catastrophic nature the costs of these floods include major economic, social and environmental aspects (Petersen, 2001). These impacts are caused by both hydro- and morpho-dynamics during discharge events. To reduce the emergency costs and enhance the preventive flood mitigation measures, understanding the forces of flow and related channel change throughout discharge events is important. Thus, so that the flood mitigation measures are possible to allocate temporally most efficiently, more understanding of river morphodynamics during discharge events is needed.

Most geomorphological studies have been concentrated on large discharge events, due to their impacts on river channel change, related river environments and human infrastructure (Greenbaum and Bergman, 2006; Grodek et al., 2012; Nardi and Rinaldi, 2015; Hooke, 2016b). However, moderate and low flows have also been shown to cause great morphological changes in gravel bed river channels (Calle et al., 2015; Hooke, 2016a), as a small discharge over a long time spans can substantially rework the sediment and fluvial bedforms produced by greater floods (Greenbaum and Bergman, 2006).

According to Hooke and Mant (2000), the pattern and magnitudes of fluvial morphological changes show the best relationship with the magnitude of peak discharge. However, during flash floods, it is difficult to perform sediment transport or bedform migration measurements to detect the timing of topographical changes, i.e. whether the greatest changes occur for example due to the peak discharge, the slope of the rising limb, or the length of the receding limb. Case studies have reported most erosion during the rising and the peak flow phases in perennial rivers (e.g. Gendaszek et al. 2013), but similar knowledge for ephemeral river channels is still limited. It is important to understand the capacities of ephemeral rivers for sediment deposition and flooding due to the combined effects of water flow and sediment transport during flood situations. If the total erosion and deposition is greater during the receding phase than the rising phase, the receding phase should not be ignored when planning flood mitigation measures.

Conventionally, ephemeral river channel changes associated with specific flood events are interpreted on the basis of the post-flood bedform and grain size distribution (e.g. Euler et al., 2017). New insight on morphodynamic changes and their driving parameters during flash floods can be gained by applying simulation methods. Simulations may provide information about the channel dynamics from the times when it has not been possible to perform measurements (Lotsari et al., 2014a) and thus increase our understanding of sediment dynamics during flood events (Hooke et al. 2005). However, morphodynamic modelling of gravel migration has so far been more common in perennial braided gravel bed rivers (e.g. Williams et al., 2016b) or laboratory flumes (Kaitna et al., 2011) than in ephemeral rivers. There are examples of simulating sediment transfer in ephemeral channels (Graf, 1996), although most of the few recent simulations have been performed in sandy reaches (Billi,

2011; Lucía et al., 2013) and on alluvial fans (Pelletier et al., 2005). Ephemeral rivers with gravel-sized bedload particles have rarely been simulated. One of the few existing examples was carried by Hooke et al. (2005), simulating morphological changes during flash floods with a cellular automata model. Their model worked well with simulations using moderately large discharges during clear water conditions, but discharge events with sediment loads had some tendency for excess deposition (Hooke et al., 2005). Further examination of different discharge events with modelling approaches is thus required.

It is possible to calibrate morphodynamic models based on the pre- and post-flood topographies of different consecutive flood situations. This causes uncertainties to the modelling, as calibration and validation data is measurable only during dry channel conditions. However, it is the best available way to get understanding on the temporal evolution of the ephemeral river channels during flooding. In particular, the differences and similarities in model performance between different flood magnitudes can be detected (Lotsari et al., 2014a; Williams et al., 2016a). Recently, the measurement techniques for deriving this calibration data for morphodynamic modelling have increased. One of them is accurate laser scanning (mobile and terrestrial), which enables the channel topography before and after flooding to be captured in detail (Milan et al., 2007; Vaaja et al., 2011; Calle et al., 2015; Kasvi et al., 2015; Kukko et al., 2015). Laser scanning enables rapid measurements of gravel bed rivers at the sub-grain level resolution (Milan et al., 2007). In ephemeral rivers the quality of topographical data can be very good and the uncertainties are less than in perennial rivers, either with laser scanning or traditional RTK-GPS measurements, because these rivers can be surveyed when the river bed is dry. The high uncertainties in topographical measurements of sub-water areas in gravel bed perennial river have been related particularly to the high bed load velocities and temporal variability of bed load (Williams et al., 2015). Mobile laser scanning, for example from vehicle or backpack, provides in an ephemeral river a practical scanning angle to survey channel banks, bar lobes and vertical surfaces (Vaaja et al., 2011; Kasvi et al., 2015; Kukko et al., 2015). Also data collection over large channel bars has been enhanced with these methods. Digital elevation models produced from airborne laser scanning (ALS) data have also been applied for detecting the geomorphic effects of different discharge events (Hauer and Habersack, 2009; Croke et al., 2013; Thompson and Croke, 2013; Nardi and Rinaldi, 2015) and for recording and calibrating sediment transport models (Rodriguez-Lloveras et al., 2015). These reveal the reorganization of the channel morphology due to flood events even over large areas with great detail (Thompson and Croke, 2013). These accurate topography data, together with RTK-GPS measurements, enable more detailed calibration of morphodynamic models than before.

This paper analyzes the evolution of a gravel bed ephemeral river channel (Rambla de la Viuda, Spain) during consecutive, moderate- (March 2013) and low-magnitude (May 2013), discharge events, by applying a morphodynamic modelling (Deflt 3D) approach. Based on the simulations, we analyze 1) the timing of river bed erosion and deposition in relation to the flow hydrograph phases during moderate- and low-magnitude discharge events, 2) the hydraulic characteristics (e.g. shear stress) explaining these channel and bedform morphodynamics, and 3) the prevailing fluvial processes, and related sediment transport routing, during these different magnitude discharge events in a gravel bed ephemeral stream. This study is based on both

accurate topographical measurements with RTK-GPS survey and mobile laser scanning, before and after each discharge event, and morphodynamic simulation (2D implementation of Delft3D). Thus, the model is calibrated with data from a moderate magnitude event, and then validated based on the consecutive low-magnitude event. With this simulation method it is possible to improve and deepen the analyses earlier made based on only the survey work (e.g. Calle et al., 2015).

**2 The study area**

Rambla de la Viuda is an ephemeral stream with a catchment area of 1523 km$^2$, located in eastern Spain (Fig. 1). The river has a braided pattern associated with a high sediment supply (Calle et al., 2015). The river is prone to flash floods, and the stream flow occurs on an average of 20 days per year. Due to the high infiltration, only intensive rains, i.e. a run-off threshold of 65 mm, cause flow in the river corridor (Segura-Beltrán and Sanchis-Ibor, 2013). This run-off threshold may vary between
10 seasons, due to soil moisture and the frequency of rains. The rain producing floods in this catchment are caused by mesoscale convective complexes (autumn and spring heavy rains), and those may be up to 100 km in diameter. These rains producing the floods are related to Atlantic fronts, which do not give much spatial variability.

The study site is located 8 km upstream of the dam of the María Cristina reservoir. The simulation area is circa 440 m long
(length along the channel) and includes an advancing gravel lobe bar front, which is the lowest 200 m of the simulation area. This lowest 200 m long reach is applied as the calibration area (Fig. 1). This area was selected essentially because it was possible to estimate the bedload sediment input with certainty (Calle et al., 2015). This is due to the gravel extraction as part of river mining activities in recent decades, which has produced a flat valley bottom on which gravel bars were deposited. Recently, new bars have been deposited by floods, and gravel lobes with a sharp front have been formed in the downstream
edge of the area. The gravel bars and lobes of the simulation area are non- or weakly-armoured (see also Table 2) and move freely during low-magnitude flows.

The two events under study occurred in spring 2013. These were caused by two rain events, which were recorded at 5 minute intervals at the precipitation and gauging station provided by the Automatic System of Hydrologic Information (SAIH-Jucar)
at Vall d'Alba (Fig. 1 and Calle et al., 2015). The first rain event (March 2013) started on 4[th] of March lasting for three days with a total of 70 mm. The second rain event (April-May 2013) started on 27[th] April and lasted four days with a total of 72 mm. These rain events caused flows that lasted 13 days (started at 12:00 on 5[th] March) and 9 days (started 14:10 on 30[th] April), respectively. The peak discharges of 23 (at 11:05 on 6[th] March) and 12.5 m$^3$ s$^{-1}$ (at 9:05 on 1[st] of May) were registered at the gauging station of Vall D'Alba, respectively. These two discharge events transported 12–41 mm sized gravel ($D_{50}$ values)
according to the measurements (see section 3.3 below), which had been performed in the areas of topographical changes. These findings have been also published at Calle et al. (2015). The movement of these gravels caused the development of the bar fronts (Calle et al., 2015).

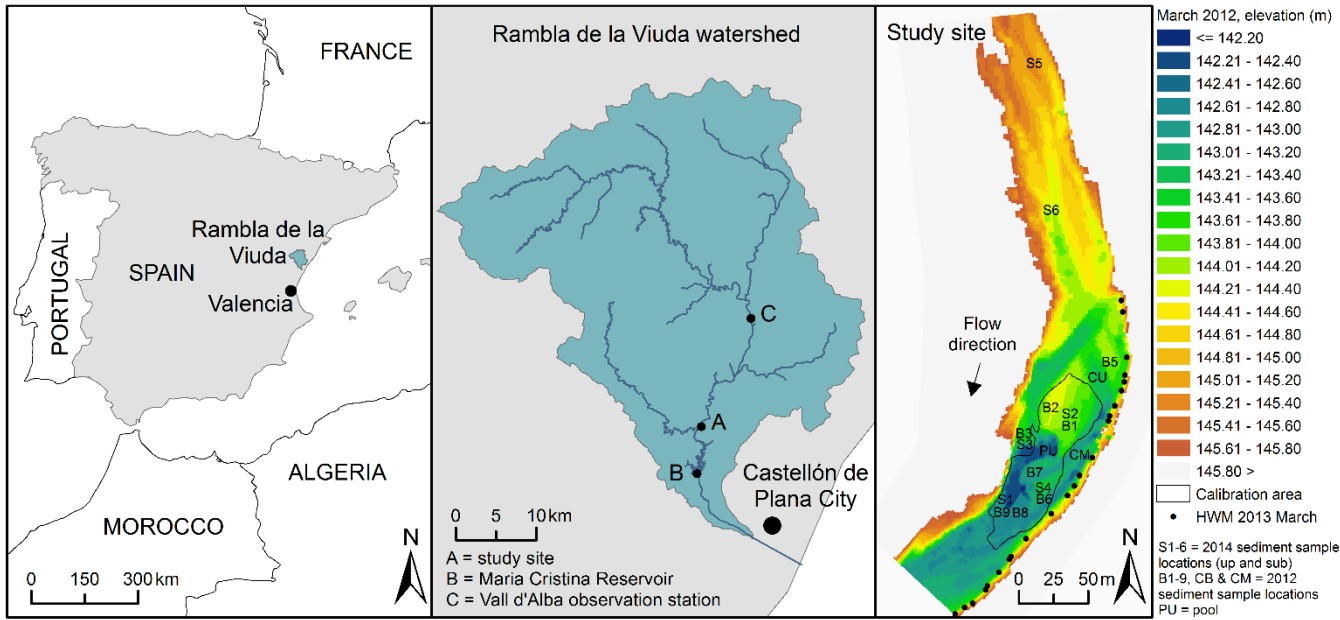

**Figure 1: The study site of the Rambla de la Viuda, Eastern Spain. The elevation on March 2012 is derived mainly from mobile laser scanning data, but the edges are from the 2009 national digital elevation model. The calibration area was defined, based on the data coverage of March 2013. The high water mark (HWM) measurement locations of 2013 March are shown (their values are plotted in Fig. 3). Also the sediment sample locations are shown (see also Table 2 and Fig. 2).**

Calle et al. (2015) described the main morphological changes in the same river reach (i.e. around the calibration area of Fig. 1) caused by these moderate (March 2013) and small (May 2013) discharge events based on multi-temporal mobile laser scanning (MLS) and RTK-GPS surveys before and after the floods (Fig. 2). They also related the observed morphological and sediment textural changes with hydraulic parameters (flow velocity, depth and discharge) estimated by a two-dimensional implementation of hydrodynamic model (Delft3D) and investigated whether the combination of the applied techniques is a reliable method to study the morphodynamics of a flood event. It was shown that MLS surveys and additional RTK-GPS surveys are suitable for a dryland river environment. In addition, a two-dimensional hydrodynamic simulation was able to estimate the hydraulic characteristics associated with the discharges. Change detection and spatial grain size distribution analyses after flood showed a high availability of material (up to $D_{84}$ of 32–45 mm) and absence of a well-developed armoured layer (Calle et al., 2015). Thresholds for sediment transport were proven to fit with the Hjulström graph (1935) in this ephemeral environment. However, simulations did not include topographical change and its influence on the hydraulic parameters during the floods, and could not answer how topographic changes evolved during rising limb, peak stage and receding limb of the hydrographs. In this paper, we investigate the ephemeral river further with morphodynamic modelling. We also answer to the above-mentioned questions and deepen the previous analyses that were based on only topographical measurements.

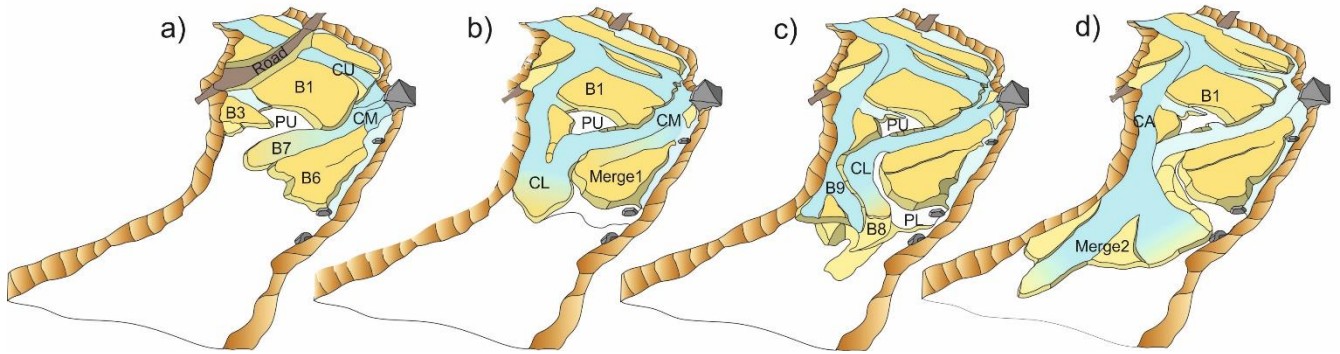

**Figure 2: The conceptual graphic presented by Calle et al. (2015: Figure 10, www.schweizerbart.de/journals/zfg): the pre-stage of 2012 (a), high flood stage during March 2013 event (b), low floor stage during May 2013 event (c) and final stage after May 2013 (d). CU: Upper Channel, CM: Middle Channel, CL: Lower Channel, PU: Upper Pool, PL: Lower Pool, B1-9: gravel Bars (cf. Calle et al., 2015).**

## 3 Morphodynamic simulation approach

### 3.1 Boundary conditions: discharges and water levels

Two flow hydrographs recorded in 2013 were defined as the upstream boundary condition and the water levels as the downstream boundary condition for the hydro- and morphodynamic model (2D module of Delft3D-flow). As no direct measurements were possible in the study reach during the discharge events, such as those demonstrated for the case of perennial rivers (Williams et al., 2013), the input data and calibration procedures differ from the traditional ones done for simulating perennial rivers. At the study reach, the hydrograph peak discharge was estimated from continuous lines of flotsam (i.e. high-water marks, HWM) emplaced by the floodwater. The hydrograph shape of the study site can be assumed to be similar to Vall d'Alba gauge station, due to the widespread continuous character of the rain events. The hydrograph shape was verified to be similar to Vall d'Alba observation station also by later water level data, which was captured with the water level sensors installed in the study site in late 2014 (see supplementary material). The HWM left by the March 2013 discharge event provided evidence that the peak discharge was greater at the study site than that measured at Vall d'Alba (Figs. 1 and 3). The hydrographs were re-scaled by using different multipliers to match the peak discharge calculated during the calibration procedure of the model (see Sect. 3.4 below, and from Calle et al., 2015).

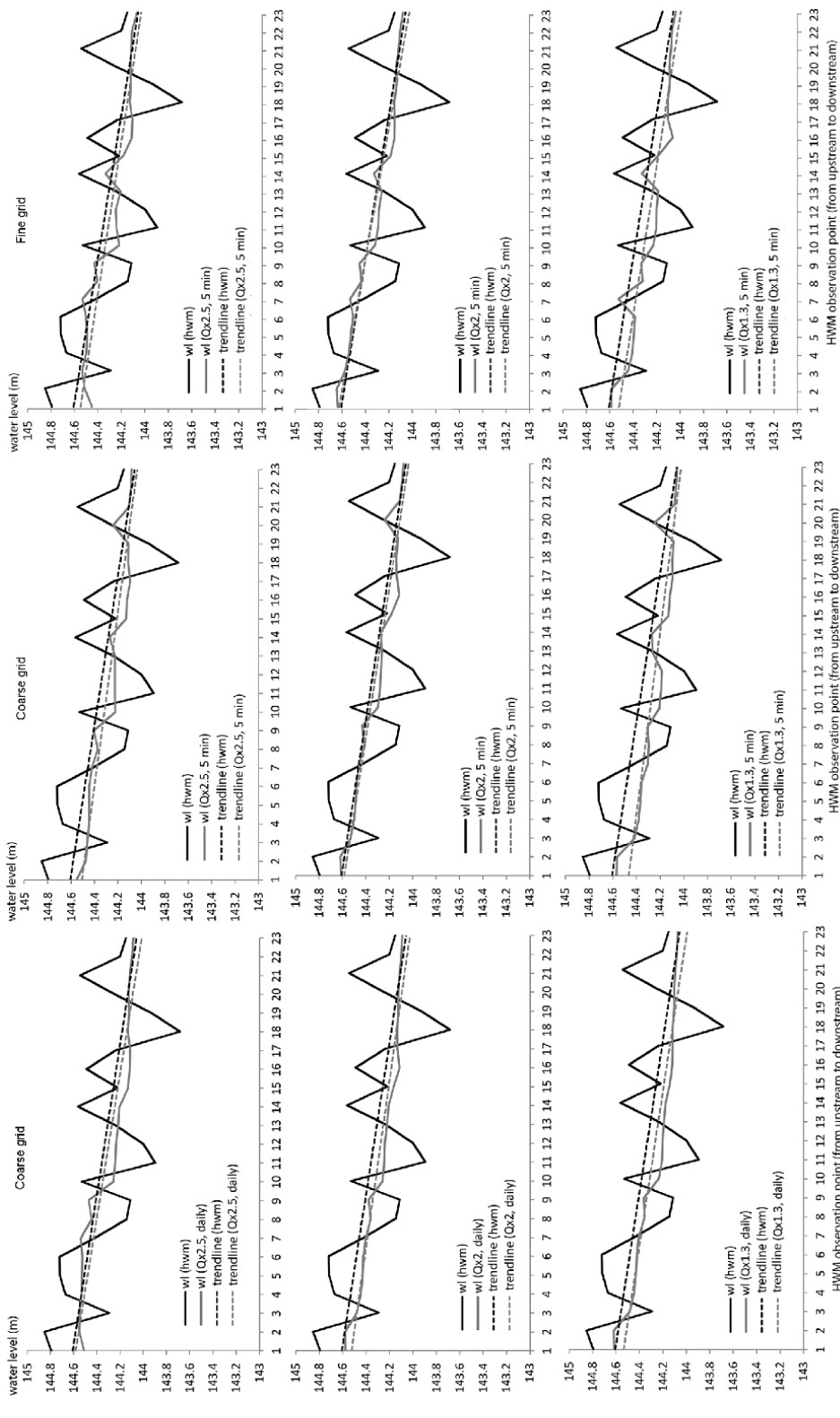

**Figure 3: The hydrodynamic calibration results based on the coarse grid (1.51–5.31 m cells) and fine grid (0.76–3.03 m cells). In this figure, the high-water marks (HWM) and simulated water levels are presented. In addition, the trendlines fitted along these measurements and simulation results are shown. These calibration results of the fine grid were previously presented in Calle et al. (2015) by the same authors. The Qx1.3, Qx2 and Qx2.5 refer to the discharges scaled to match the peak discharge at the study site. Daily and 5 min refer to the observation interval of discharges at Vall d'Alba station. wl = water level based on HWMs and simulations, trendline = the trendline fitted in the measured HWMs and the simulation results. The location of these HWM measurement points have been shown in the Fig. 1.**

Water level (hereafter WL) was defined, based on the actual HWM measurements along the longitudinal profile (Calle et al., 2015). The HWM measurements varied slightly along the longitudinal profile and the trendline, which was fitted between these field measurements, showed an accurate WL slope. Note that these HWM were measured at the valley side and it might not have reflected the water level across the whole channel as bed changes might have locally influenced water level. Therefore,

the trendline constituted a good indicative of water slope and level. The downstream-most value of this WL trendline was applied as the peak WL value of March 2013 discharge event (Fig. 3, and see also Sect. 3.4). The changes in WL over time were calculated for both March and May 2013 events based on the relative change (%) of the discharge between each discharge measurement time interval.

## 3.2 Channel geometry

The channel topography was measured, using MLS and RTK-GPS (Calle et al. 2015). Three topography data sets were applied. These were 1) initial topography (MLS, March 2012), 2) calibration topography between the floods (RTK-GPS, March 2013), 3) final validation topography after the floods (MLS, June 2013). The MLS was performed in March 2012 (pre-flood situation) and in June 2013 (after the two discharge events of spring 2013) (Calle et al., 2015 and Table 1). The MLS was performed both with a backpack and on a platform mounted on a 4x4 vehicle. A detailed description of the MLS data set can be found in

Calle et al. (2015) and Table 1. According to Calle et al. (2015) and Kukko et al. (2015) the RMSE in elevation was 18.2 mm, and it was 36 mm for the 3D position of the targets. In addition, the edges of the bar lobes, which had been moved by the March 2013 discharge event, were measured with RTK-GPS just after the discharge event in March 2013 (Trimble 4700, 1 Hz, accuracies specified by the manufacturer: XY 0.02 m, Z 0.05 m, absolute systematic measurement error $\leq$ 0.054 m). Thus, the error estimates of the topographic data can be found from Calle et al. (2015). This RTK-GPS data represented the geometry

after the March 2013 event and before the May 2013 event, and were therefore applied as a calibration data set. This enabled better estimation of the model performance compared to only using one data set after the two consecutive flood events.

Table 1. The properties of the MLS measurements of 2012 and 2013 (adapted from Calle et al. 2015).

|  | 2012 March | 2013 June |
| --- | --- | --- |
| Laser sensor / GPS sensor | FARO Photon 120 | FARO Focus 3D 120 S |
| Receiver | NovAtel DL4+ | NovAtel Flexpak6 |
| Inertial measurement unit | Honeywell HG1700 AG11 | Northrop-Grumman UIMU-LCI |
| IMU frequency | 100 Hz | 200 Hz |
| Referencing system | GPS (1 Hz) | GNSS (1 Hz) |
| Wavelength | 785 nm | 905 nm |
| Scan frequency | 61 Hz | 95 Hz |
| Point acquisition | 488 Hz | 488 Hz |
| Angular resolution | 0.045º (0.8 mrad) | 0.07º (1.2 mrad) |
| 3D RMSE | 0.034m | 0.019 m |

The 2D implementation of Delft3D morphodynamic model required channel geometry in grid format. Orthogonalized curvilinear grids of two different resolutions were created from both measured topographies, one with "coarse" 1.51–5.31 m cells and one with "fine" 0.76–3.03 m cells (i.e. circa half of the coarser grid cell sizes). These cell sizes were selected for testing the impacts of cell sizes on simulation results, but also due to their computational effectiveness. Cell sizes smaller than the "fine" resolution (i.e. 0.76–3.03 m, which is the smallest applied resolution range of the grid) did not enhance the results, and those only increased the computational time. Thus, curvilinear grids of two resolutions, "coarse" 1.51–5.31 m cells and "fine" 0.76–3.03 m cells, were created from the topography.

The initial input channel topography for all of the simulations was defined by adding the MLS 2012 measurement points to the curvilinear grids, both to the coarse and fine grids, and averaging the point values for the grid cells. There was also available a digital elevation model (DEM) from 2009 (from airborne laser scanning), which had a 1 m resolution. After adding the MLS data, this 2009 DEM was added to the grid cells, located outside the laser scanning perimeter, in order to cover the higher banks. The coarser resolution of the 2009 DEM did not affect the simulation results of the channel, because the high-water levels of the 2013 spring events barely reached these higher bank elevations.

For calibrating and validating the modelling results, topographies were also created from the March 2013 and June 2013 measurements so that those geometries had a similar resolution to the initial input channel geometry. The model outputs after each of the discharge events were then compared to these topography grids created from the March and June 2013 measurements. The grid-form geometry of March 2013, i.e. calibration topography between the floods, was created by adding only the GPS measurements to the grid cells, and the rest of the channel was excluded from the area to be analyzed during calibration (Fig. 1). The grid-form geometry of June 2013, i.e. the final validation topography after the May 2013 flood, was defined by first adding the laser scanning data set, which had been processed to include both backpack and car MLS data, to the grid cells. Then the 2009 DEM was added to the grid to cover the higher bank areas. However, only the same area, as applied in the case of March 2013 (i.e. the gravel bar lobes area), was used in the validation of the model performance. These observed elevation and volumetric changes between the events were compared to the simulated changes. For calculating the volumetric changes the curvilinear grid topographies were converted into regular grids. To minimize the errors, a 0.5 m regular grid cell size was selected, as the original cells were mostly divisible by that value.

### 3.3 Grain sizes

Spatially varying grain sizes were measured from the gravel lobes area prior to the first flood and between the floods, using the Wolman (1954) sampling method (Table 2). However, this method did not recover the differences between upper layer and sublayer sediment distribution and what kind of particles the 2013 discharge events had moved. Therefore, the gravel moved by the 2013 spring events were measured using a US standard gravelometer (US SAH-97 handheld particle analyzer) in summer 2014 (Table 2). This was possible, as no discharge had occurred between May 2013 and summer 2014. These

measurements represented different active forms, from bars to the channel bed, which had evolved during the 2013 spring floods. The bulk sieving measurements (c. 80 kg from 1.1x1.1 m area, 0–10 cm from the surface) were performed at six different locations, and from both the upper layer (UP1–6) and sublayer (SUB1–6, c. more than 10 cm below the surface) to evaluate armouring (Fig. 1 and Table 2). The upper layer-sublayer contact was established following the criteria described by Bunte and Abt (2001) considering the size of the largest particles' embedded depth. The difference in the average $D_{50}$ values of upper layer samples between 2012 (18.5 mm) and 2014 (21.1 mm) measurements was only 1.6 mm (Table 2).

Table 2. The grain sizes measured in 2012–2014 (see their locations from Fig. 1). WCM, WCU and WB1–7 were measured in 2012, WB8 and 9 were measured in March 2013, and SUB1–6 and UP 1–6 were measured in June 2014. W= Wolman, C= channel, B=Bar, SUB=sublayer, UP=upper layer. Most of the samples were within the calibration area, and their locations can be seen in Fig. 1 and also in Fig. 11 of Calle et al. (2015). Due to the spatial scarcity of the 2013 measurements (only two samples) they were not applied as the surface grain sizes for the model. The armour ratio was calculated following Lisle and Madej (1992), i.e. ratio of the surface-to-subsurface $D_{50}$.

| | upper layer | | sub layer | |
| *Sample* | $D_{50}$ *(mm)* | $D_{84}$ *(mm)* | $D_{50}$ *(mm)* | *armour ratio* |
|---|---|---|---|---|
| WCM | 18 | 29 | - | - |
| WCU | 30 | 101 | - | - |
| WB1 | 22 | 41 | - | - |
| WB2 | 17 | 33 | - | - |
| WB3 | 17 | 34 | - | - |
| WB5 | 12 | 21 | - | - |
| WB6 | 17 | 37 | - | - |
| WB7 | 14 | 22 | - | - |
| WB8 | 18 | 35 | - | - |
| WB9 | 20 | 31 | - | - |
| UP1 and SUB1 | 22.5 | 39 | 18.5 | 1.216 |
| UP2 and SUB2 | 35.5 | 79 | 21.3 | 1.667 |
| UP3 and SUB3 | 18.8 | 31 | 14.3 | 1.314 |
| UP4 and SUB4 | 18.0 | 30.5 | 14.4 | 1.250 |
| UP5 and SUB5 | 31.2 | 55.4 | 18.6 | 1.677 |
| UP6 and SUB6 | 41.2 | 80 | 16.7 | 2.467 |
| Average UP1–6 | 26.3 | - | - | - |
| Average SUB1–6 | 17.1 | - | - | - |
| Average UP and SUB1–6 | 21.1 | - | - | - |

The $D_{50}$ and $D_{84}$ grain sizes of 2012 and March 2013 were first used for the calibration and initial testing of the hydrodynamic model (see Sect. 3.4). For the calibration of the morphodynamic model, the spatial distribution of the $D_{50}$ grain sizes of both 2012 and 2014 were then applied. The grain size values were assigned to equivalent morphological elements defined in the geomorphological map. These grain sizes were transferred to each cell of the curvilinear morphodynamic model's grid. Different input grain size distributions were applied in the model tests: 1) spatially varying upper layer grain sizes, 2) spatially varying sublayer grain sizes and 3) constant average grain sizes (average of upper layer, sublayer or both upper layer and

sublayer). The values were applied to the whole active layer of the river bed. The $D_{50}$ grain size values were used in the model.

## 3.4 Hydrodynamic simulations

The same calibrated hydrodynamic model (2D implementation of the Delft3D-FLOW) was applied as in Calle et al. (2015). The flow was simulated as depth-averaged flow by using the well-known Navier-Stokes and shallow-water equations. Thus, the modelled fluid was considered vertically homogeneous. These well-known equations were found in the user manual of the model (Deltares, 2011).

Table 3.The Manning's n-values and discharges that gave the best simulation results when compared to the measured HWMs and their trend line during the calibration. Qx1.3, Qx2 and Qx2.5 stand for Vall D'Alba's discharges multiplied by 1.3, 2 and 2.5, respectively. This same table is presented in Calle et al. (2015).

| Element type | n (Qx1.3) | n (Qx2) | n (Qx2.5) |
|---|---|---|---|
| Channel/active bar | 0.063–0.065 | 0.05–0.053 | 0.03–0.04 |
| Extract area | 0.07 | 0.06 | 0.03 |
| Block | 0.068 | 0.058 | 0.05 |
| Exposed area | 0.07 | 0.06 | 0.033–0.04 |
| Scarce vegetation bar | 0.065 | 0.055 | 0.035–0.04 |
| Vegetated bar | 0.07 | 0.06 | 0.04 |
| Pleistocene terrace | 0.07 | 0.06 | 0.05 |
| Rocky bank | 0.07 | 0.06 | 0.035 |
| Vegetated bank | 0.07 | 0.06 | 0.05 |
| Opencast mine | 0.07 | 0.06 | 0.04 |

The hydrodynamic model was calibrated by adjusting the Manning's n-values so that the simulated water levels matched with the measured high-water level marks, and their WL trendline, of the 6th of March 2013 discharge peak situation (Fig. 3 and Calle et al., 2015). The hydrodynamic model was calibrated by applying first daily discharge values as the upstream boundary condition. The n-values were defined based on Limerinos (1970) equations for $D_{50}$ and $D_{84}$ grain sizes (Table 3). It is noteworthy that vegetation was almost non-existent in the channel. There was only scarce vegetation in the upstream part, and higher up in the banks, i.e. out of reach of the 2013 discharge events. Therefore, the roughness values were possible to define by the particle size on the active river channel. The Limerinos equation was applied for the whole range of water levels (i.e. hydraulic radiuses) for the March 2012 and March 2013 grain sizes of $D_{50}$ and $D_{84}$. These represented the best of the preceding conditions of the March 2013 and May 2013 flows. The Limerinos calculations were performed for a cross-section located at the downstream side of the simulation area. This cross-section was defined from 2012 geometry, i.e. pre-flood geometry, which remained unchanged during the 2013 discharge events. At the 144.151 m water level, which was the measured high-water mark elevation at this location, the wetted perimeter of the cross-section was 52.09 m and the hydraulic radius was 0.95 m (max flow depth 1.28 m). Each of the geomorphological elements, and thus also grid cells, received their own Manning's n-value during the calibration procedure. Note that the assignment of n values was static throughout the simulations, i.e. the

spatial distribution of the roughness values was not changed between simulation time steps.

The calibration was first done for the coarser grid of 1.51–5.31 m cell sizes. Because the discharge estimated at the study site was greater than the one recorded at the Vall d'Alba gauge station, three different daily discharges ("Qx1.3", "Qx2" and "Qx2.5") were tested during the calibration of the hydrodynamic model. The "Qx1.3" was defined based on the increase in watershed area (30%) between the study area and the Vall d'Alba observation station. This Qx1.3 discharge simulation matched the HWMs using the highest n-values calculated with the Limerinos 1970 equation, i.e. for shallow flow (Table 3). In this Qx1.3 discharge, each of the observations was multiplied by 1.3. The "Qx2" (observations multiplied by 2) water surface elevation matched the HWMs using an average n-values calculated from Limerinos (1970) equations. Also, the effects of bedform (from +0.01 to +0.015) and bank roughness (+0.02) had been added to these average n-values (Chow, 1959; Acrement and Schneider, 1989). The simulated water levels also matched with HWMs when the "Qx2.5" discharge and n-values representing the high-flow stage values (i.e. lower n-values on the value range) calculated from Limerinos (1970) equations were selected. Similar n-value ranges were gained for different water level situations, when either the equation for $D_{84}$ or $D_{50}$ grain sizes was applied. Based on the Limerinos equation calculations, the Qx2 was expected to work the best due to its average nature and inclusion of bedform and bank roughness effects.

When the calibration against the HWMs and their trendline was successful with these daily discharges and coarse grid, the same Manning's n-values were applied for simulations with a hydrograph with a 5-minute measurement interval. The "Qx1.3", "Qx2" and "Qx2.5" versions of these 5-minute hydrographs were tested. These simulations also reproduced the water levels, and the best fit was obtained with "Qx2" (Table 3). Thus, no more adjustments were done for the n-values. Finally, the 5 min hydrograph simulations were performed with the fine grid (0.76–3.03 m cells, Fig. 3). These simulation results corresponded to the observations better than if the coarse grid cell sizes were used (as also shown in Calle et al., 2015). Finally, both March and May 2013 discharge events were simulated. Similar multipliers (Qx1.3, Qx2, Qx2.5) and Manning values were applied for both events. Both these discharge events were simulated as unsteady flow and by applying the bathymetry derived from 2012 topographical measurements. The Qx2 discharge was proven to best produce the observed hydrodynamics (Fig. 3; see also Calle et al., 2015). All in all, 21 simulations were run during the calibration of the hydrodynamic model.

### 3.5 Morphodynamic simulations

Calibration of the morphodynamic model involved adjusting the model parameterization so that the model output geometries matched the measured bedform geometries as well as possible. The model applied an "upwind" bed update scheme, where the elevation of the bed was dynamically updated at each computational time step. In all simulations, the initial boundary conditions of the input and output sediment transport load amount were defined as 0. This is because the flood events started from 0, and no significant transport, i.e. evolution of bedforms was observed upstream and downstream of the simulation area. The model then calculated the transport based on the selected transport equations and using equilibrium concentration for

carrying input sand sediment fractions. This was similar to earlier studies done in a gravel bed river (Williams et al., 2016b), and was selected based on earlier experiences of the river (Calle et al., 2015), and because no suspended load measurements were available and the sand sediment fraction was almost non-existent in the river. Based on the experience in the study site, the load should be in equilibrium particularly in erosional areas of the study site. Because the bed level gradient affects the

bedload transport, the slope in the initial direction of the transport (referred to as the longitudinal bed slope) and the slope in the direction perpendicular to that (referred to as the transverse bed slope) were utilised. The transverse slope affects transport towards the downslope direction (Deltares, 2011). The Bagnold (1966) equation was applied for the longitudinal slope and Ikeda (1982), as presented by Van Rijn (1993), was applied for the transverse slope.

Table 4. The morphodynamic simulations and applied parameters. The fine grid size is 0.76–3.03 and the coarser size is 1.51–5.31 m. EH=Engelund-Hansen, MPM=Meyer-Peter and Müller. Events: 1=only the March 2013 event was simulated, 2=both the March and May 2013 events were simulated. The simulations that were selected for the channel change analyses are bolded.

| Simulation | Transport equation | Discharge events | transverse slope | grid size | grain size |
|---|---|---|---|---|---|
| 1 | EH | 1 | 1.5 | coarse | varying, 2012 |
| 2 | EH | 1 | 1.5 | fine | varying, 2012 |
| 3 | EH | 1 | 1.5 | coarse | varying, upper, 2014 |
| 4 | EH | 1 | 1.5 | coarse | varying, sub, 2014 |
| 5 | EH | 1 | 1.5 | coarse | varying, average upper+sub, 2014 |
| 6 | EH | 1 | 1.5 | coarse | constant, average upper, 2014 |
| 7 | EH | 1 | 1.5 | coarse | constant, average sub, 2014 |
| 8 | EH | 1 | 1.5 | coarse | constant, average upper+sub, 2014 |
| **9** | **EH** | **2** | **3** | **fine** | **varying, upper, 2014** |
| **10** | **EH** | **2** | **1.5** | **fine** | **varying, upper, 2014** |
| 11 | MPM | 2 | 1.5 | fine | varying, upper, 2014 |

Altogether, 61 morphodynamic simulations were needed for calibrating the morphodynamic model. These 61 simulations

included simulations with Qx1.3, Qx2 and Qx2.5 discharge hydrographs. During these simulations the match of the simulated water levels to HWMs was also checked. The roughness values were found to be still valid. Out of these 61 morphodynamic simulations, 11 simulations done during calibration were selected to be presented in the supplementary material of this paper (see also Table 4). The reason for their selection was that these 11 simulations showed the effects of grain size (before [2012] and after [2014] floods, and grain sizes from different layers [2014]), grid size (coarse: 1.51–5.31 m, fine: 0.76–3.03 m),

transverse slope (user defined coefficients in the bed load transport equations: default 1.5 and increased to 3) and transportation equations (Engelund-Hansen [EH], Meyer-Peter and Müller [MPM]) on model performance (Table 4, and supplementary material). These parameter tests were selected for the calibration procedure, as these had earlier been found to be important for Delft3D (2D implementation) model simulations, albeit in perennial rivers (e.g. Kasvi et al., 2014). The morphodynamic simulation results were compared to the measured topographies within the calibration area, in particular to the volumetric

change of river bed and displacement of the lobe front. Due to the better correspondence of simulations with Qx2 discharges to the observed channel evolution, all these selected morphodynamic simulations had these input data. The longitudinal slope did not affect the results, and the default value of 1.0 was selected to be used. The model was first calibrated with the March

2013 discharge event (simulations 1–11), and then validated with the May 2013 discharge event (simulations 9–11).

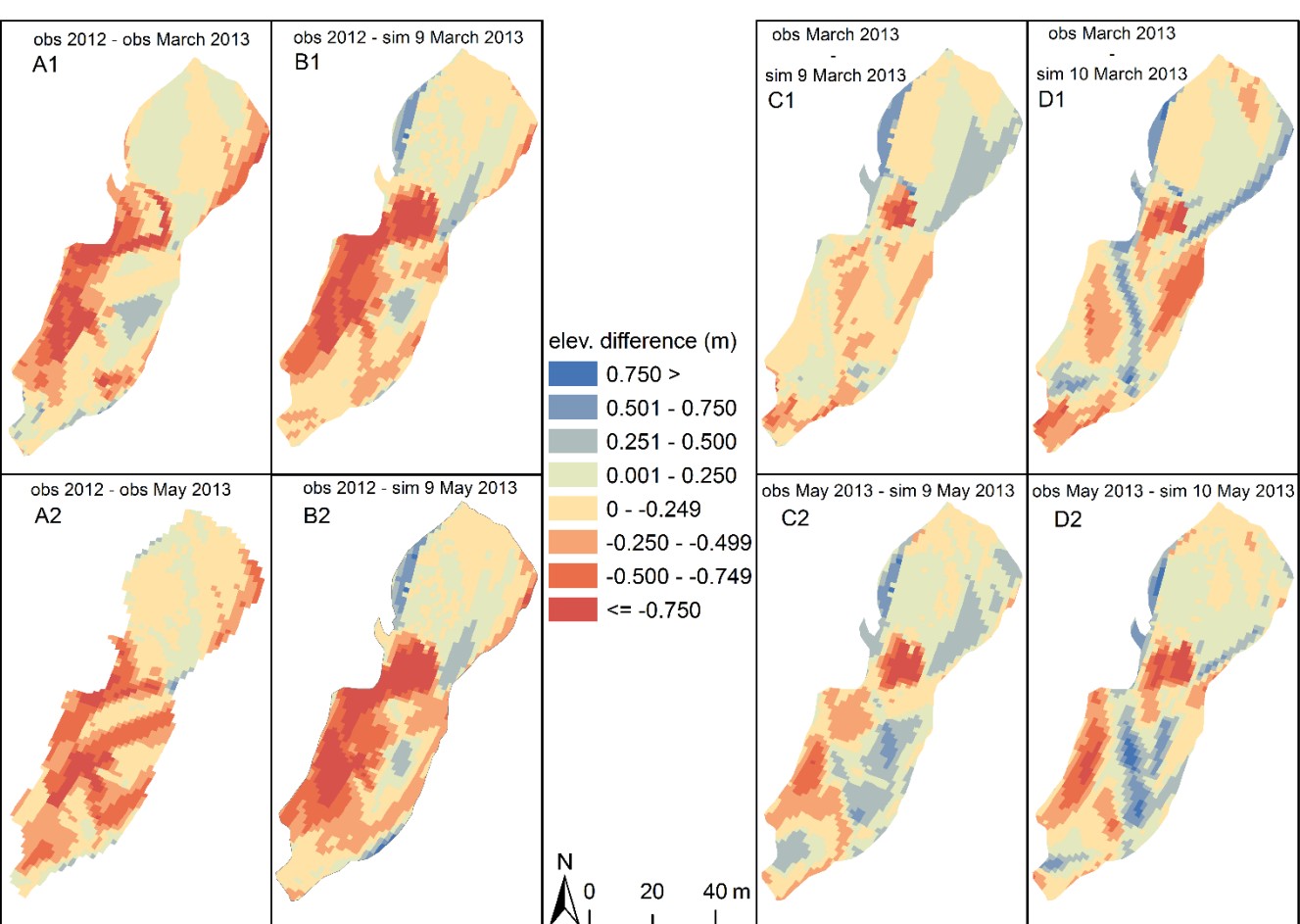

**Figure 4: The comparison between the observed and simulated (simulations 9 and 10) elevations. The negative values (red) mean that the observed (A)/simulated (B: simulation 9) March (1)/May (2) 2013 topographies were higher (i.e. deposition had occurred) than was observed in 2012. The negative values also mean that simulated elevations of March/May 2013 were higher than was observed in March/May 2013 with simulations 9 (C) and 10 (D). Simulation 9 corresponded slightly better the observations than simulation 10, and thus its results are shown in more detail. The pool area, in the middle of the area, was the only area where simulation results clearly showed more deposition than observed (see also Fig. 2 PU).**

Here we present the summary of the calibration results, which can be found in more detail in the supplementary material of this paper. The best simulation results in relation to the surveyed volumetric changes were achieved with the fine grid simulations. The coarser grid size overestimated incision and was therefore discarded. The grain size and its spatial variation affected the model results greatly and needed the largest number of tests (supplementary material). When the spatially varying upper layer sediments of 2014 (representing the sediments moved by the two discharge events) were used (simulation 10), the volumetric changes were less and fitted the observations better than if spatially varying 2012 "before floods" grain sizes were applied (simulation 2). The bedforms were also best represented when upper layer sediments were applied (simulations 9 and

10).

The deposition and erosion amounts resembled the observed elevations, and the lobe movement distances were the best when the transverse slope was increased (simulation 9). The best correspondence of the simulated and measured spatial bedform pattern was obtained with simulations number 9 and 10. However, the default (1.5) transverse slope of simulation 10 resulted in a more excavated thalweg, when compared to the simulation 9 with an increased transverse slope (Fig. 4).

Earlier simulations done for ephemeral rivers, e.g. Hooke et al (2005) and Graf (1996), had applied the Bagnold (1966) total load equation. However, the Meyer-Peter and Müller (MPM, 1948) equation had previously been proven to perform well, i.e. better than, for example, Bagnold (1980) and Parker (1990) equations in flash flood simulations and in ephemeral gravel river channels (Reid et al., 1996; Cao et al., 2010). The widely applied equations of "Engelund-Hansen 1967" (EH) and "Meyer-Peter and Müller 1948" (MPM) were selected from the standard Delft3D model's assortment of equations (cf. Deltares, 2011), because they had been developed by using initially non-armoured bed conditions and were the most appropriate for the study area, according to the $D_{50}$ grain sizes. The MPM was expected to perform the best, as it had originally been developed for particles of 0.4–29 mm overall diameter. The EH was selected as the total load equation, due to its proven performance in a variety of environments (e.g. Kasvi et al. 2014), even though it had originally been tested on up to 0.93 mm median particle sizes (Engelund and Hansen, 1967). The transport equation had a crucial role in the simulation results. The EH (simulations 9 and 10) was superior in reproducing the channel morphology. MPM resulted in much smaller and incorrect transport values (simulation 11) during both March and May 2013 discharge events.

Thus, simulations 9 and 10 produced the best correspondence with the observed riverbed changes and bed formations (Fig. 4). These were used for analysing the temporal riverbed evolution during the moderate- and low-magnitude discharge events.

## 4 The temporal evolution of Rambla de la Viuda

### 4.1 The evolution during moderate-magnitude discharge event

All in all, lobe movement of c. 60 m was observed during the March 2013 discharge event and an additional c. 30 m during the May 2013 discharge event. The bedload sediment mass balance was possible to estimate. Deposition was greater than erosion based on measurements (i.e. in the calibration area) (Table 5). Hydrographs and erosion and deposition rates were also plotted against time for the simulations 9 and 10 (Fig. 6), which had reproduced the best the observed morphodynamics within the calibration area (see Sect. 3.6). Sediment erosion and deposition were calculated from the total channel bed change amount for each hour from this calibration area. In addition, the bed elevation changes between the key time steps, i.e. times of clear changes in discharge or bed evolution, were also defined from the whole simulation area (Fig. 6: simulation 9 as an example).

Both simulations experienced more deposition than erosion during the March 2013 event (Table 5 and Fig. 5). The greatest total erosion and deposition (m$^3$) occurred at the beginning of this discharge event (Fig. 5). However, the initial changes were slightly smaller when the default transverse slope was applied (simulation 10). The peak of channel bed changes occurred at the beginning of the moderate-magnitude discharge event, i.e. before the discharge peak. This suggests to possible positive hysteresis phenomenon. Significant changes were observed in small peaks at 9h and 71h, which give an idea of the high system response to steep changes in the discharge, rather than a progressive increase (as seen in the absolute peak of 23h). The deposition in simulations 9 and 10 started very slowly decreasing after 13 hours, but the decrease in erosion was slightly faster. However, all in all, the deposition and erosion remained high for a long time and only started clearly declining after 71 hours, i.e. the secondary discharge peak. The erosion and deposition volumes followed the changes in receding discharge more faithfully than during the rising phase, occurring simultaneously to the changes in discharge. Thus, the morphodynamics of both simulations 9 and 10 followed the changes in discharge better during the falling stages of the discharge peak, i.e. when the discharge was 30 m$^3$ s$^{-1}$ or less (Fig. 5). During the rising phase, there were temporal differences between changes in discharge and changes in deposition and erosion. When the default transverse slope was applied (simulation 10), the erosion and deposition continued to be high longer in the receding phase than when the transverse slope was increased to 3 (simulation 9). The great deposition amount during the receding phase was mainly due to the propagation of the bar lobe front.

Table 5. The volumetric changes and lobe movement of observations and the two selected simulations (see also supplementary material). Gravel storage was calculated by subtracting the erosion from deposition.

| sim | grid | erosion total m$^3$ | difference compared to observed % | deposition total m$^3$ | difference compared to observed % | Gravel storage total m$^3$ | difference compared to observed % | lobe movement (m) |
|---|---|---|---|---|---|---|---|---|
| | | 1. event (March 2013) | | | | | | |
| obs | fine | 154.56 | | 984.72 | | 830.16 | | 58.5 |
| **9** | **fine** | **164.69** | +7 | **1119.28** | **+14** | 954.59 | +15 | **59.9** |
| **10** | **fine** | **178.69** | +16 | **1190.85** | **+21** | 1012.16 | +22 | **52.1** |
| | | 2. event (May 2013) | | | | | | |
| obs | fine | 63.67 | | 1087.59 | | 1023.93 | | 87.1 |
| **9** | **fine** | **182.53** | **+187** | **1222.32** | **+12** | **1039.79** | **+2** | **61.3** |
| **10** | **fine** | **160.09** | **+151** | **1289.38** | **+19** | **1129.29** | **+10** | **79.5** |

When detecting the channel changes spatially during this March 2013 moderate flow event from the whole simulation area (based on simulation 9, as an example), the channel evolved and bar lobes advanced on both sides of the large lateral bar, due to the diverted flows (B1 location of Figs. 1 and 2, and B in Fig. 6). However, the main flow and bar lobe movement occurred on the right bank side of the channel (B in Fig. 6) and diagonal bar aggradation took place in the downstream part of the study site. Both topographical observations and model simulations showed the development of the diagonal bar alongside with the cutoff of the bar (B). The model results showed that the initial cutoff and simultaneous initiation of the diagonal bar took place during the rising limb, more precisely the couple of first hours of the flood event (Fig. 6). However, during these first hours of

the flood event, the changes in the river bed were local, and greatest elevation changes occurred especially in the bar lobe area, and right upstream of the large lateral bar B1 (A location in Fig. 6). It is noteworthy that the diagonal bar would not have developed to its full extent without the long receding phase of the flood hydrograph. Spatially, the most changes occurred throughout the whole simulation area during the 24 hours following the discharge peak, i.e. between 6[th] and 7[th] of March. The diagonal bar formation was slightly greater in the model outcomes than based on observations. Despite this, the model showed potential in producing the channel development following the established theories of gravel bed evolution. Similarly to the diagonal bar movement in the downstream part of the study site (B, Fig. 6), also changes occurred in another lateral bar further upstream (A in Fig. 6). This bar experienced excavation on the right bank side, and the propagation of lobe movement downstream particularly after the discharge peak. Thus, throughout the study area the initial hours of the channel changes caused the selection of the flow and sediment transport routes where the most erosion would take place later during the moderate-magnitude discharge event.

In addition to the braiding display of the gravel bars, i.e. alternating bars and pools, the river channel has also a broader meandering planform. The simulated area can be considered as one bend. The downstream lobe area locates downstream of the bend's apex, and the greatest erosion occurred during the lowering flood phase on the right bank side, which is the inner bank of the bend at the inlet area to the bend (A). This followed also the results of other meandering river studies about the erosion locations (Lotsari et al., 2014b). Thus, during the higher flow stage the flow routing and channel started acting more like in a meandering river. Instead, during lower flow stage, such as during the initial rising flood stages, when not all of the bars were covered with water, the fluvial processes and related channel changes resembled braided channel development.

## 4.2 The evolution during low-magnitude discharge event

During the May 2013 discharge event, the hourly changes in erosion and deposition within the calibration area followed the discharge evolution more than during the March discharge event (Fig. 5). The discharge event of the May 2013 had two peaks, of which the latter discharge peak was greater. The erosion and deposition peak occurred approximately an hour after this greatest peak discharge had been reached. This suggests to possible negative hysteresis phenomenon. Thus, the deposition and erosion peaks did not occur immediately at the beginning of the discharge event, as in the case of the March 2013 event. The erosion became greater than the deposition four hours after the beginning of the discharge event (simulation 9). Six hours after the sediment transport peak (at 25 h), the deposition dominated again. During the March discharge event, the erosion was never greater than the deposition. Note also that the discharge remained constant after both peaks, whereas a decrease in the deposition and erosion rates was observed (Fig. 5).

When considering the whole simulation area (based on simulation 9, as an example), the main flow path was again on the right bank side of the channel, where the greatest changes also occurred during this low-magnitude discharge event (B in Fig. 6). The quick rise in discharge during the 1[st] hour and between the 16[th] and 19[th] hours of this discharge event did not cause spatially

great changes to the river bed, as most of the changes were within +/- 10 cm throughout the simulation area. Thus, the spatial morphodynamics of this low-magnitude event differed from the moderate-magnitude discharge event. The channel acted more like a braided river during this May 2013 low-magnitude event, as there was continuous small bed elevation changes throughout the event. Thus, there was not such clear high erosion or deposition periods, but continuous steady changes. The bar lobes progressed downstream, as the sediment was transported form the proximal side of the bars to their distal side.

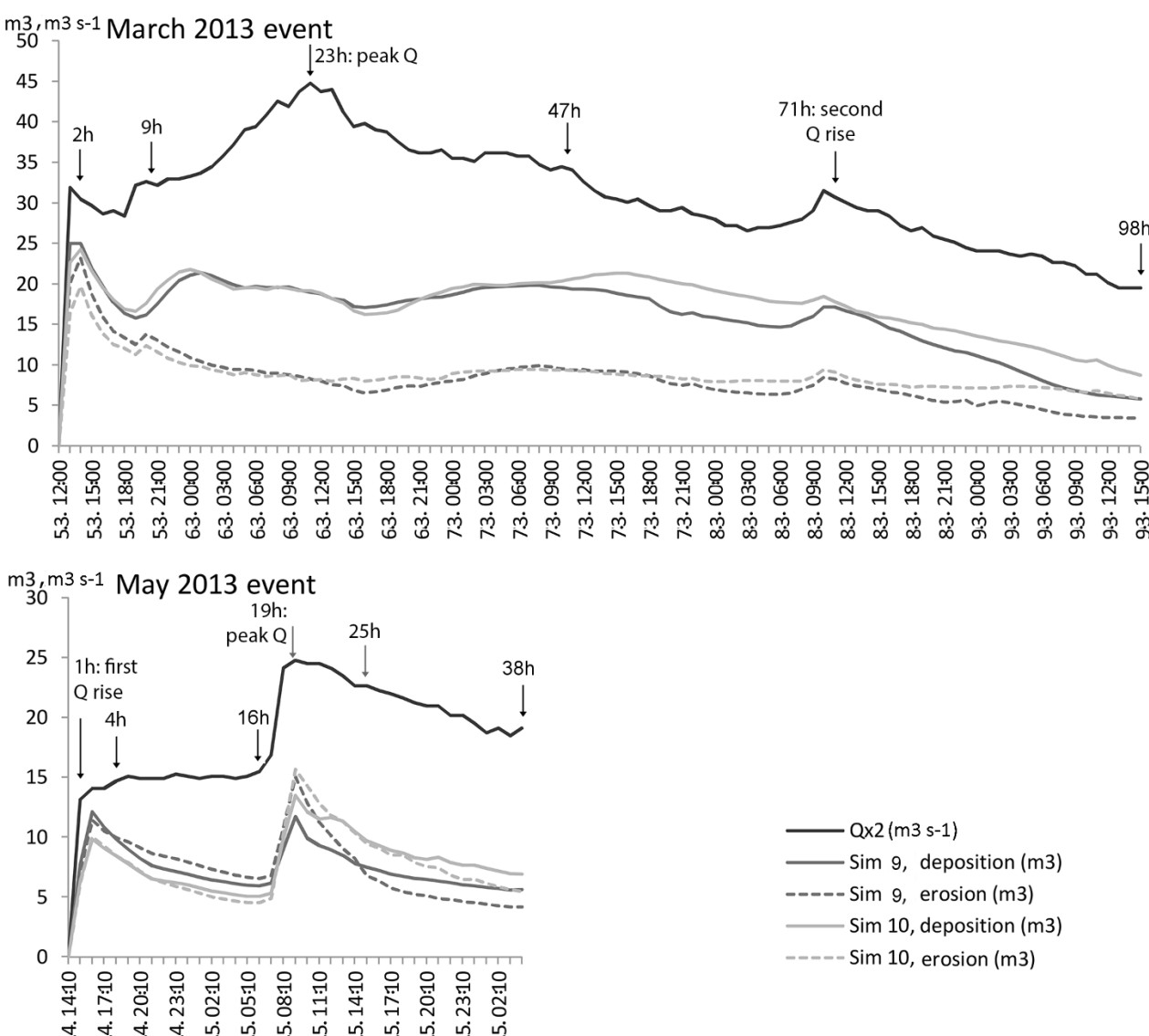

Figure 5: The hourly volumetric changes during the moderate- and low-magnitude flow events of March and May 2013. The results are based on simulations 9 and 10. The graphs show the erosion and deposition from the beginning of the flow events (12:00 on 5th March and 14:10 on 30th April) until the time when the erosion and deposition had declined and levelled out during the receding phases. The key time steps are pointed out as hours from the beginning of the flow events.

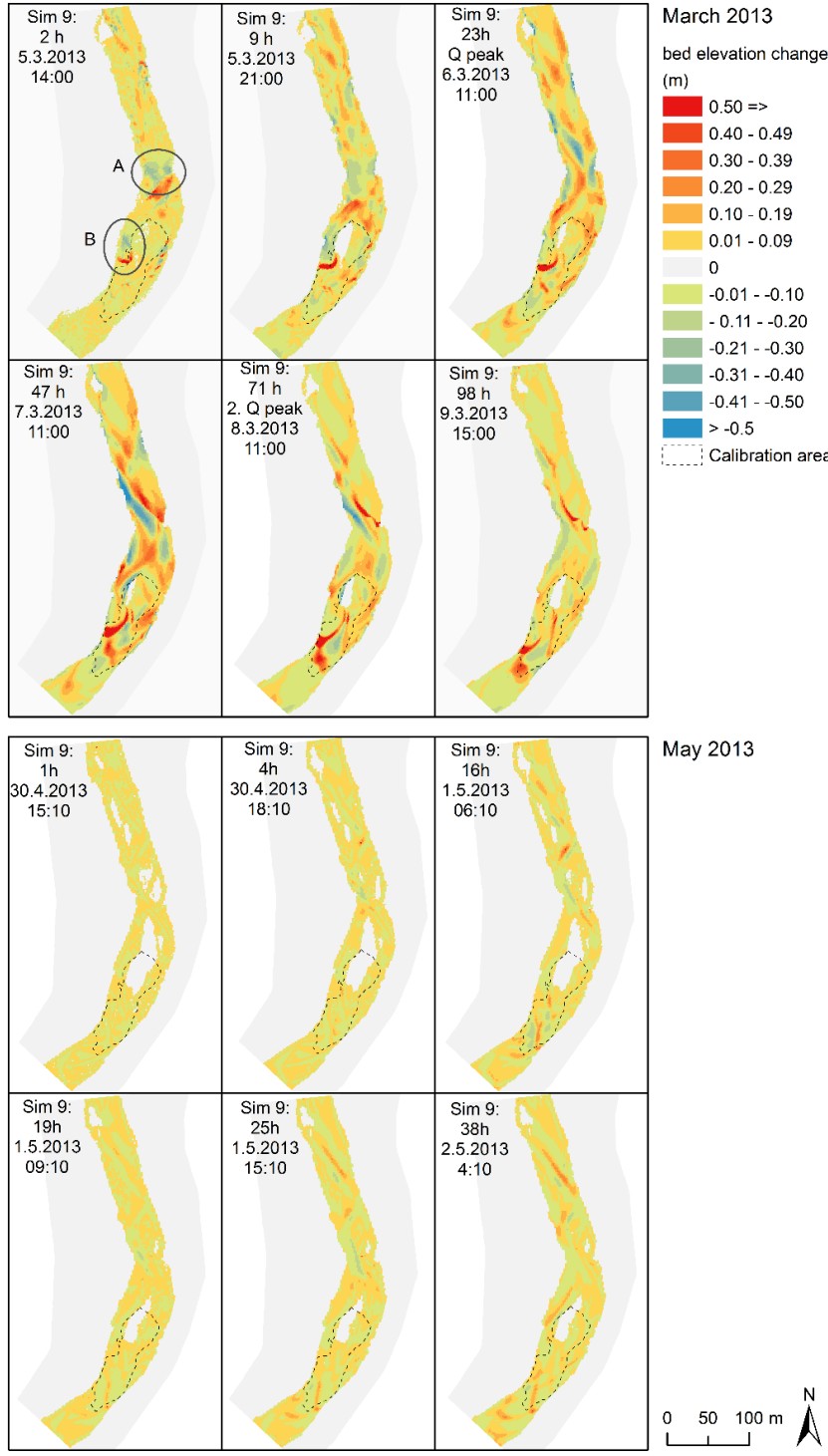

**Figure 6: The modelled bed elevation changes between the key time steps of March and May 2013 events (cf. Fig. 5). The changes of the 5th March 2017 at 14:00 are from the beginning of the discharge event.**

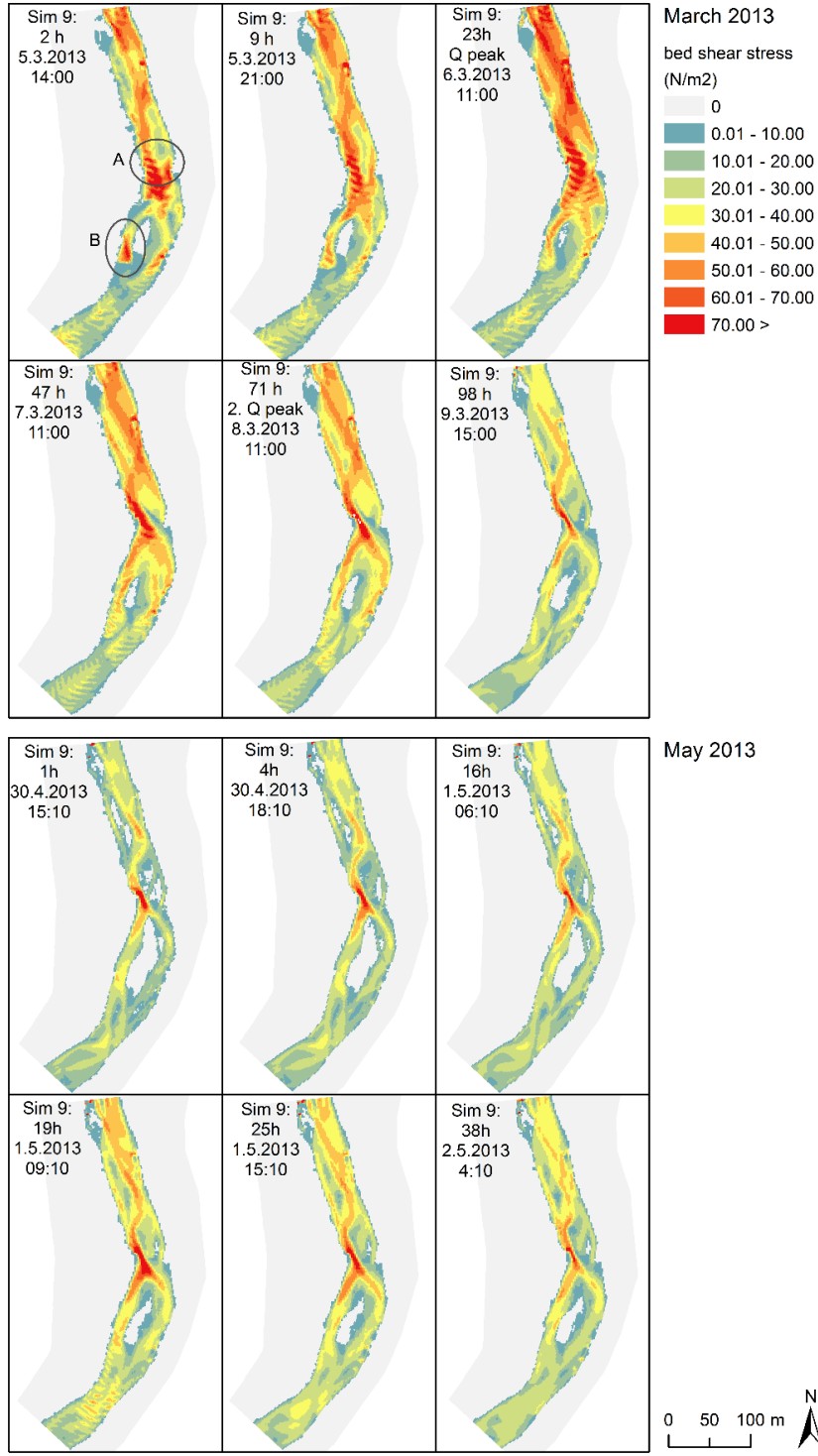

**Figure 7: The bed shear stress of simulation 9 during moderate and small discharge events. The key time steps are presented (cf. Figure 5).**

## 4.3 The flow characteristics during the moderate and small discharge events

The bed shear stress (based on simulation 9, as an example) during the first hours of the March 2013 moderate-magnitude discharge event revealed that the bend apex (A in Fig. 7) and right bank side of the downstream lateral bar (B in Fig. 7) experienced the most fluid forces. These explained the locations of the greatest changes and the initial cutoff of the lateral bar (B). It is noteworthy that these locations were the initial high erosion and bed load transport locations, but the high bed shear stresses occurred spatially more widely during the rising stage (9th hour) and the peak flow situation (23rd hour). Even though the changes in the river channel had been great between the 24 hours following the discharge peak (Fig. 6), the fluid forces (bed shear stress) had already started to concentrate on the thalwegs of the channel (Fig. 7). It is noteworthy that throughout the receding phase of the moderate-magnitude March 2013 discharge event the greatest shear stresses occurred mainly in thalwegs. Thus, the initial forces of the flood were greater throughout the channel area, but later concentrated on the channel routings formed by the initial stages of the flood discharges, particularly at right bank side at the apex (A) during the receding phase of the flood.

The spatial distribution of the bed shear stresses of the low-magnitude discharge event (May 2013) differed from that of the moderate-magnitude event (Fig. 7). Throughout the discharge event, the greatest fluid forces concentrated in the thalwegs, which had been formed by the preceding moderate-magnitude discharge event of March 2013. Particularly, it could be seen that the bed shear stresses were the greatest at the inner bank side at the apex (A), and then the main flow route was at the right bank side (B) in the lobes area. The greatest bed shear stresses occurred during the second rise of the low-magnitude discharge event, i.e. between 16 and 19 hours from the beginning of the event. Thus, similarly as revealed in Fig. 5, the transport followed the changes in discharge more during this low-magnitude May 2013 event than during the March 2013 event (Figs. 6 and 7).

## 5 Discussion

### 5.1 Uncertainties related to the simulations

The morphodynamic simulation of the ephemeral river provides a good quantification of the sequence of channel changes described by Calle et al. (2015). It also extends the analyses done based on only the topographical data, and is the only way to understand topographical changes during the flash floods. The reliability of the presently applied and also future models, which are calibrated against the events under interest, can be improved with the quality and temporal density of the available calibration topography, i.e. pre- and post-flood bedform geometries. We had two high accuracy MLS and one RTK-GPS topographical data sets, which is more than in many other studies. In recent studies done in perennial rivers, where topographical measurements have been sparse, the greatest 2D morphodynamic model uncertainties have related to the channel topographies (Sanyal, 2017). The high uncertainties in topographical measurements of sub-water areas in gravel bed perennial river have been related particularly to the high bed load velocities and temporal variability of bed load (Williams et al., 2015).

However, despite the high quality data from two events at Rambla de la Viuda, we think that further research with multiple yet-to-come events needs to be run to assess the repeatability and validation of the model even better. For example, at Rambla de la Viuda, large floods have not yet occurred since the beginning of the MLS measurement approaches. As also earlier has been stated (Verhaar et al. 2008; Lotsari et al., 2015), the roughness conditions defined for small discharge events might not be suitable for simulating extreme events. Therefore, the work and refinement of the model will continue, and the applicability of the model for larger floods will be tested, when validation data are available.

Transport rates (suspended and bed load) and flow measurements, which are measured during the events for calibrating models of perennial rivers, are always difficult and dangerous to perform in this particular ephemeral river. The uncertainties of the present model approach thus relate to the lack of sediment transport, flow and topographical data during the events. However, the selected initial boundary conditions seem to be congruent with the flooding mechanisms of Rambla de la Viuda and also with other published works, i.e. Williams et al (2016b), for loosely consolidated sand and gravel. So, the modelled flow carried each sand sediment fractions (suspended), which had been adapted to the local flow conditions at inflow boundary, and the model assumed that very little accretion or erosion was experienced near the model boundaries. Based on measurements, the presence of sand size particles and their concentration in the study site were also almost non-existent, and no channel changes occurred at downstream boundary of the simulation area. Thus, this equilibrium load condition was considered valid, and it was also the only option for the present modelling approach, as no input suspended load measurements were available. Similarly as Williams et al. (2016b) state, the model results, i.e. the modelled deposition and erosion, when compared to observations, could have been possibly enhanced if the input suspended sediment load observations would have been available. However, according to Sanyal (2017) the sediment transport is always inherently approximate in nature, and sediment load added to the model causes uncertainties to the results, despite detailed sediment load measurements have been used as model input.

Despite this lack of the data during the flood events, we had a good control on sediment volume and gravel particle-size moving downstream as the forefront lobe prograded over a flat valley bottom (as gravel bed had been mined). Total volume input and total transport rates observed in earlier study of Rambla de la Viuda by Calle et al. (2015), had already proved the high availability of sorted gravel particles, and were the basis of the decisions made while building up the model. As already mentioned, the channel changes at the output downstream boundary of the studied reach was zero. In addition, the simulation result supports the hypothesis of Calle et al. (2015) that moderate- and low-magnitude events reworked sediment locally within the reach. This means that these flows were not able to establish a sediment connection upstream, i.e. between larger reaches, and the transported sediment originated from the erosion of adjacent areas.

In addition, we were able to use a discharge and precipitation stations further upstream of the study area, and thus also a hydrograph with a known shape (cf. supplementary material). Sometimes only the high water marks are known, and for

example, Cao et al. (2010) analyzed the bedload transport by using a symmetrical hydrograph. This could be why their simulations showed less transport during the receding phase than ours. An et al (2017a, 2017b) and Viparelli et al. (2011) had raised the issue related to the cyclic hydrographs and hydrograph boundary layer (HBL), which defines the spatial region within which riverbed topography and grain size respond to the hydrograph. They state that the HBL causes problems, if the material is poorly sorted. The calibration area of the present study was in the downstream part of the simulation area, i.e. outside of the influence of the possible HBL. The gravel of Rambla de la Viuda was well sorted and also therefore we can state that the HBL was not an issue for the calibration of our model.

As the curvilinear grids of fine 0.76–3.03 m and coarse 1.51–5.01 m cells were tested for morphodynamic simulations, the finer grid was proven to better predict the channel evolution, including both flow and bar lobe movement paths. Grids of circa 3 m cell sizes have been found by Williams et al. (2016b) to be suitable for Deflt3D model applications. Furthermore, the grain sizes and their spatial distribution affected the morphodynamic simulation results greatly. Hooke et al. (2005) have shown that grain sizes have a major effect on the morphological impacts of floods. Simulation results may be enhanced when using techniques that are able to obtain spatially more complete and detailed grain size data (e.g. Wang et al., 2013) from the surface layers, instead of lower resolution site sampling such as Wolman method. Earlier studies have stated that a calculation of the bedload transport rate of mixtures should be based on the availability of each size range in the surface layer (Parker, 1990), and that during the calibration of Deflt3D model uniform sediment cannot be an acceptable assumption to study the long-term response of natural gravel-bed braided rivers (Singh et al., 2017). The best simulation results of Rambla de la Viuda were indeed gained when spatially varying upper surface layer sediment sizes were applied. However, further analyses of spatial and vertical grain size distribution effects on ephemeral gravel-bed river simulations are suggested. The next step should be to apply the surface grain size variation derived from the laser scanning data to analyse these effects of grain size distribution in further detail (Casas et al., 2010a and b; Kukko et al., 2015).

The patterns of channels, bars and braiding properties are highly sensitive to the bed slope parameterization, and particularly transverse slope affects the sediment transport in long-term simulations (Schuurman et al., 2013; Kasvi et al., 2014; Singh et al., 2017). Also during these rather short-term flood event simulations at gravel bed Rambla de la Viuda, the effects of the increasing transverse slope on channel evolution was evident, and the simulation results resembled more the observed topographies. Kasvi et al. (2014) had noticed that the higher the transverse bed slope effect (i.e. value 3) was, the more filling of pools occurred in a meandering river. Similar to Kasvi et al. (2014), the pool area of Rambla de la Viuda experienced more deposition with the increased transverse slope value (Fig. 4). This pool area was the only clearly wrongly simulated location, when this high transverse slope value was applied, but otherwise the lobe movement and elevation best resembled the observations. Thus, in a gravel bed river with short lasting flow events, adjustments are clearly needed in this parameter in order to replicate the observed changes while simulating with the currently available sediment transport equations. Thus, similar to the perennial river simulations (e.g. Kasvi et al., 2014), the transverse slope was important for the simulated changes.

Despite the MPM equation having earlier been successfully applied in ephemeral gravel bed rivers (Reid et al., 1996), the movement was minimal in Rambla de la Viuda and occurred only in the beginning of the moderate event. Therefore, after these tests with the MPM equation, it was discarded from the final analyses. The transport rate is imposed as bed-load transport due to currents in both EH and MPM equations, which are used in Delft3D-FLOW. The MPM is more advanced than EH as the MPM includes a critical shear stress for transport (Deltares, 2011). However, this could be one reason for less movement, if critical stresses for transport required in MPM equation were not exceeded during the lower discharges of the hydrographs. According to Barry et al. (2004) the formulae containing a transport threshold typically exhibit poor performance when compared to the observed gravel bed river's bed load data. These equations include MPM (1948) and Bagnold (1960) equations, which have been applied in ephemeral channels. In addition, Singh et al. (2017) state that during variable discharge conditions, the MPM formula can not produce the braided river pattern because MPM is not suitable if the value of the Shields parameter falls outside its applicability range. These above mentioned facts could explain, why the MPM equation did not move particles during the lower discharges and during the receding phase of the hydrographs of Rambla de la Viuda, where flow is temporally highly variable. The application of the MPM equation could also partly explain why the receding phase's transport was less in the study of Cao et al. (2010), than in the present study. Despite being unable to produce the correct morphodynamics caused by the moderate- or low-discharge events of Rambla de la Viuda, the MPM equation could have potential in the simulation of greater floods of ephemeral gravel bed rivers, similarly as Reid et al. (1996) and Cao et al. (2010) have shown. However, further research on this matter should be done also with higher discharges at the river reach.

Engelund and Hansen (1967) developed their equation, so that the effects of dunes, e.g. transport from stream-side to lee-side and form drag, were included. This could be the reason why the EH worked the best at the Rambla de la Viuda, where clear sharp lobe edges occur, and the local slope varies a lot. In addition, the EH is a total load equation (Engelund and Hansen, 1967), whereas, for example, the MPM equation calculates bedload. There are no universal sediment transport formulas, which can cover all coarse sediment transport conditions from low discharges to peak discharges (Li et al. 2016; Singh et al., 2017). The grain sizes, which had been moved by the 2013 discharge events, were on the upper limit of the available sediment transport equations. Also a matter of further research is to detect, whether the transport equations would able to move the sediments at higher discharges, if in reality the higher discharges would possibly transport even larger sediment particles.

However, despite the challenges of applying morphodynamic simulation approaches to ephemeral river channel change analyses, the results are promising, and show the importance of both rising and receding phases. The observations of sediment transport during flow hydrographs would still be important, in addition to the continuous time series of pre- and post-flood topography, but at the moment, their measurement is still a complex and risky issue. By continuing the long-term observations of the ephemeral channel changes with the high-precision equipment, more reliable calibration and validation of the models may be gained.

## 5.2 Moderate- and low-magnitude flow events as channel modifiers

Our study has shown that geomorphic responses to the two analysed discharge events differed. Previously this uniqueness of geomorphic response has been shown for perennial rivers (Pitlick, 1993), where also topographical and sedimentary data has been applied as initial conditions. Hooke (2016a) stated that the flow events of similar magnitude can have differing effects, depending on the state of the system, as the long-term evolution of the ephemeral river channel and its material greatly influence the response to the stream flow. Some events are more erosional and some are more depositional (Hooke, 2016a). Moreover, Hooke et al. (2005) noted the importance of simulating and analysing the feedback effects of consecutive events. At Rambla de la Viuda, the riverbed morphology formed by the March 2013 event influenced the later channel changes during the May 2013 flow. For example, the flow was diverted to the right bank side during the first flood (of March 2013), which also therefore acted as the main channel for the May 2013 flow, which was lower in magnitude.

The simulation results of the Rambla de la Viuda showed that the differences between rising, peak and receding phases of a moderate-magnitude discharge event are very important in an ephemeral river environment. A higher total amount of channel changes occurred during the receding phase than at the early stages of the discharge events. Deposition dominated due to the progradation of the frontal bar lobe, particularly on the right bank side of the channel. Thus, the continuous channel changes were similar to those for braided perennial rivers (Lotsari et al., 2014a). However, the channel changes differed from a recent study of Gendaszek et al. (2013), who studied the gravel perennial riverbed changes during moderate- ($65 \, \text{m}^3 \, \text{s}^{-1}$) and high-flow ($159 \, \text{m}^3 \, \text{s}^{-1}$) events. They found that most erosion occurred during the rising and the peak flow phases, but did not mention great changes during the receding phase. They found only some scour during sustained high flows following the flood peak (Gendaszek et al., 2013). Noteworthy is that they applied one sensor per reach, and thus the site selection could have greatly affected on their results.

Ferguson (1993) stated the potential in numerical modelling of the coupling between geometry, flow, and bedload transport, if it can be applied successfully to braided channels. The results were promising at Rambla de la Viuda, which also has a braided pattern. According to Wheaton et al. (2013) the chute cutoff mechanism, already described by Ferguson (1993), is the most common braiding mechanism, but that the cutoff is not only an erosional process, but more the result of deposition during the construction of diagonal bars. In our study area, there was a situation resembling to chute cutoff, as the channel was cut more on the right bank side of the bar than before. The modelling was capable in producing this observed chute cutoff from the right bank side during the moderate discharge event (March 2013). The high values of bed shear stresses related to this initial chute cut off (B location in Figs. 6 and 7). Both erosion and deposition related to these changes, as both topographical observations and model simulations showed the development of the diagonal bar alongside with the cutoff of the bar. However, the diagonal bar formation was slightly greater in the model outcomes than based on observations. Despite this, the model showed potential in producing the channel development following the established theories of gravel bed evolution. In addition,

the model showed that the initial cutoff and simultaneous initiation of the diagonal bar took place during the rising limb, but the diagonal bar would not have developed to its full extent without the long receding phase of the flood hydrograph. Williams et al. (2015) had found that the choking is the main process for braiding development in their studied perennial river. However, we were not able to observe choking processes within our study reach. Further analyses of this braiding process are needed to perform from longer river reaches of ephemeral rivers.

In addition to these braiding processes, the fluvial processes at higher discharges resembled more the ones in a meandering river bend. The greatest erosion and bed shear stresses occurred on the right/inner bank side at the inlet area of the bend (A in Figs. 6 and 7) during the peak and receding flood phases of the moderate-magnitude (March 2013) event. This followed also the results of the erosion and high velocity core locations of perennial meandering rivers (Dietrich and Smith, 1983; Lotsari et al., 2014b). However, during lower flow stage, such as during the initial rise of the floods and the whole May 2013 discharge event, the spatial distribution of channel changes resembled more to braided channel development. Therefore, the morphodynamics of the low-magnitude May 2013 event differed from the preceding moderate-magnitude discharge event. During the May 2013 low-magnitude event, there was continuous steady small bed elevation changes throughout the event. It is noteworthy that throughout the receding phase of the moderate-magnitude March 2013 discharge event and the whole May 2013 event, the greatest shear stresses occurred mainly at the channel routings formed by the initial stages of the moderate-magnitude flood discharges.

It is noteworthy that during the moderate-discharge event the erosion and deposition peak occurred much earlier than the discharge peak. However, the low-magnitude discharge event experienced the greatest channel changes an hour after the discharge peak. Our model results (e.g. Figs. 7 and 8) suggest the possible existence of hysteresis in the rate of bedform changes, being positive in the case of moderate-magnitude flow (bedform change peak occurs before flood peak), and negative in the case of low-magnitude flow (bedform change peak occurs during/after flood peak). The hysteresis phenomenon has been described well in sediment transport studies, and their effect is due, among other factors, to sediment depletion or surface gravel consolidation in the channel (Reid et al., 1985), or a long-lasting portion of the baseflow during the recession limb (Walling, 1974). Cao et al. (2010) have shown, based on their 1D simulations, that bedload transport in an ephemeral river can have similarities to a perennial river. However, even though perennial rivers may have sharp rising phases in their discharge hydrographs (e.g. Long, 2009), they more likely have a greater initial threshold for particle movement by bed-armouring than ephemeral rivers (Reid et al., 1996; Hassan et al., 2009). Although, the armouring of perennial rivers can decrease as bed load concentrations increase (Müller and Pitlick, 2013). Hysteresis of both kinds have also been shown in perennial rivers regarding turbidity, but their flashy storm hydrographs have more often caused anti-clockwise (i.e. negative) hysteresis phenomenon (Lloyd et al., 2016). This would indicate that ephemeral rivers act more similarly to perennial rivers during their low magnitude flow events. Even though further research is needed, the results indicate that the greater the discharge event's magnitude in an ephemeral river is, the more different the channel evolution and its timing are compared to perennial braided gravel bed rivers.

In our study reach, 30 $m^3$ $s^{-1}$ was the threshold discharge for the channel changes to become gradual and to follow the discharge changes during the receding phase. According to Hooke (2016a), the threshold values of hydraulic conditions for erosion and deposition in ephemeral channels vary between sites, mostly due to the size and nature of the bed material. Threshold discharge

for the deposition and erosion of 15 mm particles has been observed to be 35 $m^3$ $s^{-1}$ and 1 $m^3$ $s^{-1}$, respectively, and of 50 mm particles 14 $m^3$ $s^{-1}$ and 40 $m^3$ $s^{-1}$, respectively (Hooke, 2016a). The average upper and sublayer $D_{50}$ grain sizes at the Rambla de la Viuda were 26.3 mm and 17.1 mm, respectively, and the maximum $D_{50}$ grain size was 40.2 mm. The discharges of the Rambla de la Viuda were overall within the ranges of Hooke's thresholds for the movement of these-sized particles, and the simulations clearly showed their transport. In addition, the threshold discharge of 30 $m^3$ $s^{-1}$ for starting the channel changes of

the Rambla de la Viuda to follow the discharge changes during the receding phase coincide well with Hooke's (2016a) threshold analysis of deposition. However, the threshold could be also site dependent, such as for Spanish Mediterranean region, where both Hooke et al. (2016a) and the present study have been performed.

## 6 Conclusions

A 2D morphodynamic model was successfully implemented for inferring the morphological changes caused by moderate- and

low magnitude discharge events. The morphological starting and final conditions were derived from high-resolution laser scanning topography, bedform mapping and sediment texture analysis of both surface and subsurface layers on the main morphological bars and channel units. The flow hydrograph was based on a continuous (5 min) record from an upstream gauge station, which was re-scaled to the peak flow estimated at the study reach based on flotsam evidences. The model parameters were calibrated to best fit the pre- and post- topographic field evidence. The following conclusions can be made about the

morpho- and hydrodynamics in an ephemeral river during moderate- and low-magnitude discharge events:

**1)** The 2D implementation of morphodynamic model was proven to work during moderate- and small-magnitude flash flood events in an ephemeral river. The spatially varying grain size data, the applied transverse slope parameter value, and the streamflow hydrograph were the most important factors affecting the simulation results of the bedform and channel evolution. The selection of sediment transport equations was also critical to match the sediment mass-balance, concluding that the total

load equation (Engelund-Hansen, EH), worked the best. When modelling events of moderate and low magnitude in ephemeral rivers of 50 to 100 meters wide, a resolution of at least 1m is recommended (i.e. one cell is 1–2 % of the total width), because it is able to show the steep lobes forefront scarp.

**2)** Morphodynamic models can shed light on the channel evolution during flash flood events. Both rising and receding phases of discharge events were predicted to be important for bar movement and channel evolution and thus should not be ignored

while planning flood mitigation measures. The erosion and deposition can be greater during the long-lasting receding phase than at the rising phase of moderate- and low-magnitude discharge events, despite the typical hydrograph shape of a flash flood. The receding phase contributed also greatly on the shaping of the bed forms and channel pattern.

**3)** The deposition and erosion peak rates were predicted to occur at the beginning of the moderate-magnitude discharge events (e.g. March 2013), whereas deposition dominated throughout the event, i.e., even during the rising phase. On the contrary, the low-magnitude discharge event (e.g. May 2013) only experienced the greatest channel changes after the discharge peak.

**4)** These different predicted erosion/deposition patterns suggest that the timing of the channel change peak, compared to the timing of the discharge peak, differs between different magnitude discharge events. These stress the importance of previous flood history (timing, succession and magnitude) in understanding the geomorphic response of gravel bed ephemeral rivers. The peak of channel changes during the moderate flood occurred before the discharge peak. On the contrary, the low-magnitude discharge event experienced the peak of channel changes after the discharge peak, suggesting a rupture of upper layer before sediment was moved. These differences between the events were at least evident with these two events and the applied parameterization. However, further studies of this possible hysteresis effect are needed from multiple discharge events.

**5)** The results showed that the fluid forces of the initial stages of the moderate-magnitude discharge event caused the initiation of chute cutoff and diagonal bar formation, and defined the flow routing of the rest of the moderate magnitude discharge event. The flow during the following low-magnitude discharge event followed these same thalwegs caused by the preceding moderate-magnitude event.

**6)** The clearest difference in the predicted riverbed changes between the rising and receding phases were that erosion and deposition characteristics followed the temporal discharge changes more during the receding phase than during the rising phase. The threshold discharge, below which the channel-bed changes started following the discharge changes temporally, was around 30 $m^3 s^{-1}$. Noteworthy is that this ephemeral channel acted like the braided river channel during these lower flow conditions, but when the bars were submerged in higher discharges the high fluid forces followed the meandering river planform of Rambla de la Viuda. Thus, there were different fluvial processes working at different flow stages, particularly during the moderate-magnitude discharge event.

**7)** Preventive measures against flash flood-induced channel changes should take into account moderate- and low-magnitude flows. This has received support from earlier studies (e.g. Hooke, 2016a and b) as the receding phase can cause substantial channel changes, it plays a major role in inducing damages/modifications to the river environments, and it needs to be acknowledged while planning flood mitigation measures.

## Data availability

The data has been stored in the universities of the authors. We have provided the initial geometry, calibration area delineation, sedimentary data (2014 samples) and final topography from the calibration area as supplementary material. Because some of the data was gained from third parties, we are not able to provide the whole model as open access data. Data can be also accessed by request from the authors. Each request will be processed separately.

## Author contribution

Lotsari has done most of the writing and all of the model simulations. Calle and Benito have been also greatly contributed in the writing of the paper and have enabled the study with their projects, as they have initiated the studies of Rambla de la Viuda. All the other authors have contributed to the writing process by commenting the manuscript and its content. All authors have contributed in collecting the field data. Laser scanning has been done by Kukko, Kaartinen, and Alho. The laser scanning data has been processed by Kukko and Kaartinen. Hyyppä J. and Hyyppä H. and have contributed by providing funding for the study and they have also worked on the development of the laser scanning approaches. Benito, Calle and Lotsari have measured the sedimentological data. GPS measurements of topography and high water marks have been done by Benito and Calle.

**Competing interests**

The authors declare that they have no conflict of interest.

**Acknowledgements**

We would like to thank the Institut Cartografic Valencia, Servicio Automatico de Informacion Hidrologica of the Confederacion Hidrográfica del Júcar for supporting data. Financial support is provided by the Spanish "Ministerio de Economia y Competitividad" (projects CGL2011-29176 and CGL2014-58127-C3-1-R), the Academy of Finland (Extreme and annual fluvial processes in river dynamics – ExRIVER [grant number 267345], the Centre of Excellence in Laser Scanning Research – CoE-LaSR [grant number 272195], and the Strategic Research Council project "COMBAT" [grant number 293389]), Maj and Tor Nessling Foundation (Äärimmäisten ja vuotuisten fluviaaliprosessien vaikutukset jokidynamiikkaan [grant number 2013067]), and the Spanish Ministry of Economy and Competitiveness through the projects PALEOMED (CGL2014-58127-C3-1-R) and EPHIMED (CGL2017-86839-C3-1-R). Finally, we would like to thank the editors and the three reviewers for comprehensive reviews, which greatly helped us to improve the manuscript.

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
