# Peer review of "Topographical change caused by moderate and small floods in a gravel bed ephemeral river - a depth-averaged morphodynamic simulation approach"

_Earth Surface Dynamics, 2017_

## Referee Comment (RC1) · Anonymous Referee #1 · 9 Oct 2017

This manuscript describes the calibration of a depth-averaged Delft3D morphodynamic model for an ephemeral gravel bed river. Subsequent analysis of model predictions, for a small and moderate flood, examine the relative importance of rising and falling hydrograph limbs for erosion and deposition. Pre- and post-event topographic surveys, acquired using mobile terrestrial laser scanning and RTK-GNSS, are used to provide topographic data to provide boundary conditions and evaluate model performance. The use of a spatially-distributed morphodynamic model to gain insight into ephemeral river processes is novel, promising and interesting. I think the manuscript has the potential to make a significant contribution. However, I have some major concerns:

[Figure]

1) The morphological model calibration exercise is not especially novel (see e.g. Schuurman et al., 2013; Williams et al., 2016b). However, the application and analysis of the morphodynamics of a braided ephemeral river is, in my opinion, where this manuscript has the greatest potential for impact. I would recommend that the paper is re-focused (in the introduction, results, discussion, conclusions) to emphasise the geomorphological findings that arise from the morphodynamic model results (as analysed in section 5.2 which considers flood sequencing, rising/falling hydrograph, diagonal bar formation, hysteresis and discharge thresholds). This would distinguish this paper from other published work that has discussed and explained the relative importance of representing different processes in morphodynamic models of gravel-bed rivers.

2) The morphological model description does not mention bank erosion. Is bank erosion represented in the model? Bank erosion has been identified as an important process in perennial gravel-bed rivers (Wheaton et al., 2013). Although existing, simplified bank erosion algorithms (e.g. repose schema in Delft3D) may be inadequate in representing natural rates of lateral adjustment (Stecca et al., 2017) there is a need to justify the incorporation (or not) of lateral adjustment in the model.

3) The study area description describes how a lobe is forming in the model domain? Is this a result of enhanced sediment supply from upstream? Is an equilibrium sediment supply boundary at the upstream end of the model a suitable assumption?

4) Has any analysis of uncertainty in topographic been incorporated into the volumetric calculations of erosion and deposition (see e.g. Wheaton et al., 2010). Although the mobile laser scanning error is quotes as 0.034 m for the March 2012 survey (Table 1) this may still be significant depending upon the magnitude of changes, particularly deposition. In addition, it would be useful to have figures showing patterns of observed and predicted morphological change. I think these would be more useful for evaluating model performance than the maps of observed and post-event predicted topography (within section 4.2).

5) Spatially variable roughness is used for some simulations (description P11 L15). For morphodynamic simulations, is roughness recalculated from the surface grain size distribution or is spatial variation fixed a priori?

6) The modelling undertaken is calibration rather than a sensitivity analysis (P13 L2).

7) The presentation of methods and results on the MPM bedload transport formula. On P14 it is stated that this is not integrated into the Delft3D code (which it is, see Williams et al., 2016b) yet this is contradicted on P19 where MPM results are discussed.

8) I am not convinced that the threshold of 30 cumecs is widely applicable (conclusion 7). What about the influence of other river styles, sediment sizes, vegetation interactions)?

9) Some sentences require sharpening / clearer expression (some are identified below). Check methods and results are consistently written in the past tense.

10) Data availability statement: since this is an open access journal I would expect the observed topography and model input files (as a minimum for the calibrated model runs for the small and moderate events) to be packaged up and available for download (perhaps with a dataset doi – see e.g. re3data.org to search suitable repositories). This would promote the "reproducibility" of the research and enable readers to re-run the models, if they were interested in analysing aspects of the results that are not presented in this paper.

OTHER COMMENTS Title: Does changes need to be plural? "Gravelly" or "gravel bed" (also consider elsewhere) Delft2D: Delft3D is the name of the software but the simulations are executed in shallow water mode. Use "depth-averaged Delft3D" or similar phrasing rather than Delft2D.

P1 L15: Change performed to caused.

P1 L18: We pursue is an odd phrase in this context.

P1 L20-22: This is the key research question – see my comment 1 above about emphasising this throughout the manuscript.

P1 L28-31: Is there sufficient evidence from the two simulated events to extrapolate a conclusion about flood sequencing?

P2 L19: Yes, dynamics during high flows are difficult to measure but there are some attempts to do this e.g. Williams et al., 2015.

P2 L23: Williams et al. citation should be 2016b not 2013

P2 L28-29: How did the Hooke et al. (2005) model perform? Be more critical / analytical in the literature review examination (and this also goes for other parts of the introduction e.g. P2 L31 – why do the uncertainties arise?)

P2 L32: References to the use of repeat surveys for morphodynamic modelling could be stated e.g. Lotsari et al., 2013; Williams et al., 2016a

P3 L2: Broader references are needed here e.g. Milan et al., 2009

P3 L1: Within this paragraph you could emphasize more strongly that since ephemeral rivers can be surveyed whilst the river bed is dry, the topographic survey is characterised by lower errors because wet areas are usually associated with greater survey uncertainty.

P3 L15: I suggest emphasising the geomorphological research questions rather than the calibration exercise (see comment 1 above).

P4 L17: Where is the evidence that these sized grains were moved?

P5 L8: Are morphological and topographic both needed?

P5 L15: Clarify whether the grain size distributions were spatially distributed.

P6 L12: Clarify in the text whether the re-scaling with linear.

Figure 3 legend: More commentary is needed. Abbreviations need to be stated. A

location map would be useful.

P8 L2: Sentence 2 – The three datasets need to be introduced before this sentence.

P8 L18-19: Sentence not clear

P8 L25: Typo MSL

P9 L16-20: Sentences not clear.

P9 L28: What guidance was used to sample a 10 cm layer?

P9 L30: Is this difference for the upper or sub layer?

P10: How many size fractions were used in the model?

P10 L11: It would be useful to refer to a map here.

P11 L1: Insert "solving" after "for".

P11 L5: This event has previously been referred to as March 2013 rather than 6.3.2013. Be consistent

P12 L12: Quantify "sufficiently well"

Table 6: Separate volumes and

Fig 4 / 5: Be consistent in use of "After" in figures (top left corner) but not in other sub-figures.

P20 top paragraph: The morphological change description here is interesting and I think the results / discussion would benefit from more analysis of this style. Section 5.1: This discussion needs to be more closely integrated into the discussion in other, similar Delft3D (or other graded sediment morphological model) calibration findings.

P25 L11: "Unique" – from sample of 2?

P25 L31: "differs" – explain why

REFERENCES

Lotsari, E., D. Wainwright, G. D. Corner, P. Alho, and J. Käyhkö (2013), Surveyed and modelled one-year morphodynamics in the braided lower Tana River, Hydrological Processes, 28(4), 2685-2716. doi: 10.1002/hyp.9750.

Milan, D. J., G. L. Heritage, and D. Hetherington (2007), Application of a 3D laser scanner in the assessment of erosion and deposition volumes and channel change in a proglacial river, Earth Surface Processes and Landforms, 32(11), 1657-1674. doi: 10.1002/esp.1592. Stecca, G., R. Measures, and D. M. Hicks (2017), A framework for the analysis of noncohesive bank erosion algorithms in morphodynamic modeling, Water Resources Research, doi: 10.1002/2017WR020756.

Schuurman, F., W. A. Marra, and M. G. Kleinhans (2013), Physics-based modeling of large braided sand-bed rivers: bar pattern formation, dynamics and sensitivity, Journal of Geophysical Research: Earth Surface, 118(4), 2509-2527. doi: 10.1002/2013jf002896.

Wheaton, J. M., J. Brasington, S. E. Darby, and D. A. Sear (2010), Accounting for uncertainty in DEMs from repeat topographic surveys: improved sediment budgets, Earth Surface Processes and Landforms, 35(2), 136-156. doi: 10.1002/esp.1886.

Wheaton, J. M., J. Brasington, S. E. Darby, A. Kasprak, D. Sear, and D. Vericat (2013), Morphodynamic signatures of braiding mechanisms as expressed through change in sediment storage in a gravel-bed river, Journal of Geophysical Research: Earth Surface, 118(2), 759-779. doi: 10.1002/jgrf.20060.

Williams, R. D., C. R. Rennie, J. Brasington, D. M. Hicks, and D. Vericat (2015), Within-event spatially distributed bed material transport: linking apparent bedload velocity to morphological change, Journal of Geophysical Research: Earth Surface, 120(3), 604-622. doi: 10.1002/2014JF003346.

Williams, R. D., J. Brasington, and D. M. Hicks (2016a), Numerical Modelling of Braided

River Morphodynamics: Review and Future Challenges, Geography Compass, 10(3), 102-127. doi: 10.1111/gec3.12260

Williams, R. D., R. Measures, D. M. Hicks, and J. Brasington (2016b), Assessment of a numerical model to reproduce event-scale erosion and deposition distributions in a braided river, Water Resources Research, 52(8), 6621-6642. doi: 10.1002/2015WR018491.
* * *

---

## Referee Comment (RC2) · Anonymous Referee #2 · 26 Oct 2017

The paper provides a detailed account of the calibration of a numerical model (a 2d implementation of Delft 3D) in order to simulate the geomorphic changes during flash flood events. The main novelty and significant findings of the work are the predictions of when sediment transport occurs during flood events of different magnitudes, based on a model simulation calibrated versus observed changes. It is an interesting and thorough piece of research.

However, the purpose of the work and the findings and conclusions get lost in a highly detailed description of the calibration method. The calibration itself is not particularly novel, although as it includes observations spanning two events it is more robust than

many other examples. Much of this detail could be moved to supplementary materials, and a concise summary provided in the main manuscript. This would allow the manuscript to be focussed more on the simulations themselves, what they show, and why this is important (I felt this last point was not made clearly enough throughout).

The authors at several points refer to sensitivity testing performed on the model. Although evidently there were some tests performed to assess model behaviour differences with some variations, these are not sufficiently thorough to be considered a sensitivity analysis, and falls short of the level of analysis performed by operators in other modelling fields. For example, the tests performed would not ascertain any non-linear interactions between the parameters being tested. For interest, Ziliani et al (2013) provides a useful and efficient methodology for screening model parameter sensitivity in reach-scale geomorphic models, which might be useful for future work. The authors should be careful using terms such as sensitivity analyses, and the conclusions they draw from the tests should not be presented with as much certainty as they presently are. The tests may be useful for informing the calibration process, but cannot be relied upon to suggest anything concrete about the model behaviour.

Below are some notes for correction or additional clarification:

Page 1, Line 15 – use "caused by the flood" or "resulting from the flood" instead of "performed by"

Page 1, Line 16 (and throughout) – strictly speaking, a 2D implementation of Delft 3D

Page 1, Line 24 – As explained above, I don't think a rigorous enough sensitivity test has been performed to make these conclusions with such certainty

Page 1, Line 24-25 – Is the total load equation probabilistic? Or is it implemented in a deterministic way too?

Page 2, Line 13-16 – I'd like to see something here to explain why this is important. Why does it matter that we know this?

Page 2, Line 30-34 – The authors are also in danger of overparamaterising the model by using a single calibration against a small set of observations. It risks equifinality with the model matching the data well, but there are many ways that the changes could have come about.

Page 3, Line 19 – With two events the authors could have calibrated with one and validated with the other as an extra check.

Page 4, Figure 1 – It would be more useful to see the reach in context with the wider catchment here than its rough location in Spain. How big is the catchment and where is the reach in relation to the gauging station used?

Page 6, Lines 10-16 – What is happening between in 18km between the gauge and the reach to increase the discharge. Are there more channels flowing in? Has there been more rainfall inbetween? Convective events are likely not to span 18km so rainfall will not be uniform. This is key as the authors make the assumption that the hydrograph is the same shape when flow makes it to the reach, just scaled larger, yet the reasons for the increased discharge will also likely change the shape of the hydrograph and this potentially invalidates the calibration and the conclusions. More detail needs to provided here to justify the above assumption.

Page 6, Lines 17-22 – By calibrating the discharge against water levels there is another assumption that the bed levels were static and had no influence on the water levels in the flood (despite in the discussion stating that bed changes are an important influencer on water levels). This will lead to some uncertainty cascading to the simulations.

Page 9, Line 30-31 – What is the expected level of uncertainty in the measurements – is 1.6mm difference significant?

Page 11, Line 15-16 – If the Manning's n values were set according the geomorphological elements, did they alter with changes during the model runs, or were they stationary?

Page 11 Line 18-20 and Table3 – either present the values in the table in the order of Qx1.3, Qx2 and Qx2.5, or present the values in the text in the order of the table.

Page 12, Line 31-32 – Tells you something about model sensitivity to parameters, but not a great deal.

Page 13, Line 1 – Need to know what the criteria was for this! "best and most interesting" not a rigorous methodology. The authors might have rejected useful information by arbitrarily labelling it as not interesting.

Page 13, Line 2 – Not a sensitivity analysis.

Figures 4, 5 and 6 – These would be easier to interpret if they showed the changes over the whole reach, not just in the area used for the analysis. This could still be identified as in Figures 1 and 8.

Page 19, Line 18 – "satisfactory" not "satisfying"

Page 19, Line 25 – or other factors had an influence? Interaction with other parameters?

Page 22, Line 8 – Has it been shown that it is reliable? It's shown it is able to reproduce the events it was calibrated against, but not others.

Page 23, Line 2 – Yes, validation of the calibrated model versus independent events is needed to claim the model is reliable.

Page 23, Line 4 – Yes, the uncertainties need to acknowledged, maybe a little more prominently then there are at present.

Page 23, Line 10 – The authors have not provided enough detail to evidence that the hydrograph is of "a known shape". It may well be, but the gauge is 18km away with half the discharge (according to the calibrations).

Page 24, Line 13 – Both equations are implemented determinisitically are they not?

Page 25, Line 13 – Are the parameter sets non-stationary or is it the initial conditions which need to be set for each event?

Page 27, Line 18 – Instead of "goodness" use "feasibility"

Page 28, Line 6 – Both implemented deterministically?

Page 28, Line 8 – This is a key point, and should be made more prominent earlier in the manuscript as part of the reason why this research is important.

Page 28, Line 16-18 – Can this be said with such certainty, or does it just apply to this reach for these events (and also with these parameters)

Ziliani et al (2013) - doi:10.1002/jgrf.20154

Thank you for an interesting study and manuscript, I look forward to seeing the revised version.

---

## Referee Comment (RC3) · Anonymous Referee #3 · 28 Nov 2017

Review of 'Topographical changes caused by moderate and small floods in gravelly ephemeral river – 2D morphodynamic simulation approach'

Earth Surface Dynamics – Lotsari et al., 2017

Overview This paper uses a 2D morphodynamic model to assess the impact of small and moderate floods on the evolution of ephemeral rivers. This is an interesting topic which will be of use to the wider community however in its present form I do not think it is suitable for publication. The authors spend over half of the paper describing the model set up and calibration and do not really address the original question. Given a lot of the model inputs were from a previously published paper a lot of the rather dense

description could be cut from the paper to allow more time for a detailed analysis of the impacts of flood characteristics. There is also little time devoted to discussing the applicability of this model to scenarios other than the very detailed description in the paper for which there is good input boundary data. Detailed comments and queries are below which the authors needs to address if this paper is to be published.

Page 2 Line 4 – I would query the word greatest and authors should consider an alternative Page 2 – Line 19-21 – three papers might be worth reviewing – although not on the context of ephemeral rivers they give useful context Viparelli et al (2011) 'A model to predict the evolution of a gravel bed river under an imposed cyclic hydrograph and its application to the Trinity River' WRR An et al (2017) 'Gravel-bed river evolution in earthquake-prone regions subject to cycled hydrographs and repeated sediment pulses' ESPL An et al (2017) 'Effect of grain sorting on gravel bed river evolution subject to cycled hydrographs: Bed load sheets and breakdown of the hydrograph boundary layer' JGR ES Page 2 Line 30 – when you say between and after flood topographies do you mean pre and post flood topographies? Page 3 Line 12 – use alternative phrasing for high/ large floods Page 3 Line 12/13 – the sentence beginning in addition does not make grammatical sense Page 3 Line 25/ 26 – consider 'The river has a braided pattern associated with a high sediment supply' instead of current wording Page 4 Line 15 – how far away was the gauging station from the study site? Page 4 Line 15 – you say the discharges at the field site were estimated to be higher but how much? How did you estimate this? Page 6 Lines 9-15 - you have assumed that the discharge between the gauging station and the reach is increasing but that the hydrograph shape remains the same but how is this so? This has important implications for the validity of the calibration of your model. Much more detail is needed to justify this assumption Page 6 Lines 17-22 – you have assumed that bed level has not changed when you have calibrated discharge to water levels – how valid is this assumption? Figure 3- it is unclear how this relates – more detailed description needed Page 8 Lines 15 – 21 better justification is needed of cell sizes – e.eg what do you mean 'did not make more difference to the results'? Page 8 Line 25 – MSL or

MLS? Page 9 – if the water level did not reach the high bank elevations why add that 2009 DEM results to the model – what does it add? Page 9 Line 16 -17 – what do you mean ' the capabilities of the model to result correct channel bed elevations'? This sentence needs restructuring. Page 9 lines 16- 20 – the meaning of this section of text is unclear Page 9 Line 30 – is this the difference between the armour and sub surface layer? What was the difference in the D84? Page 13 – what do you mean 'best and most interesting results'? This surely needs justification? What do you deem best or most interesting? Page 13 Lines 2-3 define better performance? Figures 4-6 – it would have been useful to show the 'analysed area in context with the broader area studied Page 19 line 4 – ok so how many model runs are now relevant? Page 19 Line 19 – should be satisfactory Page 19 Lines 6 – 19 – would these plots have been better as hysteresis type plots so plotting Q against volumetric changes in bedload? You have not discussed hysteresis at all? The same comments apply for section 4.2.2 Figure 7 – you need axis labels, especially for the Y axis- what is it showing? Page 22 – what do you mean reliable? How useful is it for predicting other events/ scenarios? Page 23 Lines 1-4 – you definitely need to discuss the applicability of this model to other events a and need to discuss the limitations of this approach!!! Page 23 Line 10 – did you really know the hydrograph shape? Page 25 Lines 17-18 – have you considered how the role of the changing surface structure could be incorporated into the model as this has been shown to have significant impacts in transport rates?

Please also note the supplement to this comment:
https://www.earth-surf-dynam-discuss.net/esurf-2017-52/esurf-2017-52-RC3-supplement.pdf

---

## Editor Comment (EC1) · R. Hodge (Editor) · 29 Nov 2017

Apologies for the confusion around the timing of the discussion period closing. All three reviewers agree that this paper contains some novel and interesting material that is potentially suitable for publication. However, they also all express reservations about aspects of the paper, which suggest that substantial revisions are necessary.

The reviewers are agreed that the novelty of the paper is in the modelling of a small ephemeral channel. However, the current focus of the paper is mainly on the model calibration. The reviewers felt that there had already been fairly extensive work on calibrating Delft3D and similar models, and that this part of the paper could make

more reference to previous material and be reduced in length. Rather than covering all aspects of model calibration, you could use this section of the paper to explain any aspects of this particular channel and/or ephemeral rivers in general that mean that a standard calibration approach is not applicable. Make it clearer what you have done that is different to standard calibration approaches. Be careful about the use of the phrase 'sensitivity analysis'; the work that you present is not strictly a sensitivity analysis. In line with the reviewers' comments, you want to think about how you might develop the parts of this paper that present the model application. There are some interesting ideas about the magnitude and timing of morphological changes during different sized events that could be explored further.

All reviewers also provide useful comments on other aspects of the paper that should be taken into consideration. I look forward to receiving the revised version of this paper.

---

## Author Comment (AC1) · 7 Dec 2017

analysis of model predictions, for a small and moderate flood, examine the relative importance of rising and falling hydrograph limbs for erosion and deposition. Pre- and post-event topographic surveys, acquired using mobile terrestrial laser scanning and RTK-GNSS, are used to provide topographic data to provide boundary conditions and evaluate model performance. The use of a spatially-distributed morphodynamic model to gain insight into ephemeral river processes is novel, promising and interesting. I think the manuscript has the potential to make a significant contribution. However, I have some major concerns: 1) The morphological model calibration exercise is not

especially novel (see e.g. Schuurman et al., 2013; Williams et al., 2016b). However, the application and analysis of the morphodynamics of a braided ephemeral river is, in my opinion, where this Author responses to referee (1, 2 and 3) comments

(See also the same responses from the pdf document included in the supplementary material)

RC = comments from referees

AC = author's response and changes in manuscript

(The page and line numbers refer to the revised manuscript where changes have been accepted [see supplementary material]. See supplementary material also for the "track changes" version of the manuscript.)
* * *
REFEREE #1:

RC (Referee Comment): This manuscript describes the calibration of a depth-averaged Delft3D morphodynamic model for an ephemeral gravel bed river. Subsequent manuscript has the greatest potential for impact. I would recommend that the paper is re-focused (in the introduction, results, discussion, conclusions) to emphasise the geo-morphological findings that arise from the morphodynamic model results (as analysed in section 5.2 which considers flood sequencing, rising/falling hydrograph, diagonal bar formation, hysteresis and discharge thresholds). This would distinguish this paper from other published work that has discussed and explained the relative importance of representing different processes in morphodynamic models of gravel-bed rivers.

AC (Author Comment): Thank you for these suggestions. We have modified the manuscript accordingly and refocused the paper in introduction (also aims), results, discussion and conclusions sections to emphasize the geomorphological findings. The paper now aims at analyzing (P 3 line 30 onwards) "the evolution of a gravel bed ephemeral river channel (Rambla de la Viuda, Spain) during consecutive, moderate-

(March 2013) and low-magnitude (May 2013), discharge events, by applying a mor-phodynamic modelling (Deflt 3D) approach. Based on the simulations, we analyze 1) the timing of river bed erosion and deposition in relation to the flow hydrograph phases during moderate- and low-magnitude discharge events, 2) the hydraulic characteristics (e.g. shear stress) explaining these channel and bedform morphodynamics, and 3) the prevailing fluvial processes, and related sediment transport routing, during these different magnitude discharge events in a gravel bed ephemeral stream. Understand-ing of these processes would be needed particularly for river management works and flood mitigation purposes." Along with these changes, we have modified the results section so that we have moved the morphodynamic model's calibration results to the supplementary material (as the second reviewer suggested), and present only their summary in methods section. We have added more results related to the morphody-namics (channel bed elevation changes and bed shear stresses) so that we were able to refocus the paper. Because other referees suggested more detailed discussion also related to the modelling, we have also modified the discussion section related to the modelling uncertainties.

RC: 2) The morphological model description does not mention bank erosion. Is bank erosion represented in the model? Bank erosion has been identified as an important process in perennial gravel-bed rivers (Wheaton et al., 2013). Although existing, sim-plified bank erosion algorithms (e.g. repose schema in Delft3D) may be inadequate in representing natural rates of lateral adjustment (Stecca et al., 2017) there is a need to justify the incorporation (or not) of lateral adjustment in the model.

AC: No lateral erosion algorithm was included in this commercial model version. Based on the laser scanning data, we could see that the channel changes were vertical and not lateral during these medium and small magnitude discharge events. The banks of the channel are bedrock, and the moderate- and low-magnitude discharge events were not able to erode the banks.

RC: 3) The study area description describes how a lobe is forming in the model domain? Is this a result of enhanced sediment supply from upstream? Is an equilibrium sediment supply boundary at the upstream end of the model a suitable assumption?

AC: We consider an equilibrium sediment supply as a valid assumption. The initial boundary conditions of the input and output sediment transport load amount were defined as 0. This is because the flood events started from 0, and no significant transport, i.e. evolution of bedforms was observed upstream and downstream of the simulation area. The model then calculated the transport based on the selected transport equations and using equilibrium concentration for carrying input sand sediment fractions. The selected initial boundary conditions seem to be congruent with the flooding mechanisms of Rambla de la Viuda and also with other published works, i.e. Williams et al (2016b), for loosely consolidated sand and gravel. So, the modelled flow carried each sand sediment fractions (suspended) adapted to the local flow conditions at inflow boundary, and the model assumed that very little accretion or erosion was experienced near the model boundaries. Based on measurements, the presence of sand size particles and their concentration in the study site were also almost non-existent, and no channel changes occurred at downstream boundary of the simulation area. Thus, this equilibrium load condition was considered valid, and it was also the only option for the present modelling approach, as no input suspended load measurements were available. Similarly as Williams et al. (2016b) state, the model results, i.e. the modelled deposition and erosion, when compared to observations, could have been possibly enhanced if the input suspended sediment load observations would have been available.

Despite this lack of the data during the flood events, we had a good control on sediment volume and gravel particle-size moving downstream as the forefront lobe prograded over a flat valley bottom (as gravel bed had been mined). Total volume input and total transport rates observed in earlier study of Rambla de la Viuda by Calle et al. (2015), had already proved the high availability of sorted gravel particles, and were the basis of the decisions made while building up the model. As already mentioned, the channel changes at the output downstream boundary of the studied reach was zero. In addition,

**ESurfD**
the simulation result supports the hypothesis of Calle et al. (2015) that moderate- and low-magnitude events reworked sediment locally within the reach. This means that these flows were not able to establish a sediment connection upstream, i.e. between larger reaches, and the transported sediment originated from the erosion of adjacent areas.

We have described these also in the discussion P 23 lines 10-30 and methods P 12 line 30 - P13 line 4.

RC: 4) Has any analysis of uncertainty in topographic been incorporated into the volumetric calculations of erosion and deposition (see e.g. Wheaton et al., 2010). Although the mobile laser scanning error is quotes as 0.034 m for the March 2012 survey (Table 1) this may still be significant depending upon the magnitude of changes, particularly deposition. In addition, it would be useful to have figures showing patterns of observed and predicted morphological change. I think these would be more useful for evaluating model performance than the maps of observed and post-event predicted topography (within section 4.2).

AC: The error estimates of measured changes had already been estimated in Calle et al. (2015) based on the all available data we had. Therefore, the analyses were not repeated in this study. Based on also the referee #3's comment, i.e. that methods should not be repeated and referencing to earlier studies should be done, we decided not to add more error analyses into this present paper, but we are referring to Calle et al. (2015) (P8 lines 19-20). We had already included comparison between observed and predicted morphological changes from the fine grid simulations, which were better than the coarse grid simulations (moved now as Fig. 3 of the supplementary material of calibration results, however, the same results of simulation 9 and 10 can also be seen from the new Fig. 4 of the main manuscript). We decided to modify the new Fig. 4 of the manuscript so that it now also shows the changes between observed 2012 and observed 2013 topographies, and also change between observed 2012 and simulated (simulation 9) 2013 topographies, in addition to the previous analyses of simulations 9

and 10. In addition, the predicted temporal bed elevation changes can be seen now from the new Fig. 7.

RC: 5) Spatially variable roughness is used for some simulations (description P11 L15). For morphodynamic simulations, is roughness recalculated from the surface grain size distribution or is spatial variation fixed a priori?

AC: The model version was not able to change/update the spatial distribution of roughness values during the simulations. The initial input values were applied. (P11 line 25 – P12 line 1)

RC: 6) The modelling undertaken is calibration rather than a sensitivity analysis (P13 L2).

AC: Thank you for this notification. We have changed the words "sensitivity analysis" to "calibration" throughout the manuscript and the supplementary material.

RC: 7) The presentation of methods and results on the MPM bedload transport formula. On P14 it is stated that this is not integrated into the Delft3D code (which it is, see Williams et al., 2016b) yet this is contradicted on P19 where MPM results are discussed.

AC: We meant that the Bagnold equation was not included in the standard set of equations of the commercial model we had. The MPM was all the time integrated in the model version of ours. During the shortening of the methods section, we ended up deleting the sentence in question.

RC: 8) I am not convinced that the threshold of 30 cumecs is widely applicable (conclusion 7). What about the influence of other river styles, sediment sizes, vegetation interactions)?

AC: We have modified the former conclusion 7 (now conclusions 6) and the discussion 5.2 section so that we are not anymore suggesting this limit to be applicable widely in other areas, and we now just mention that this 30 m3/s limit was observed in Rambla

de la Viuda. Thank you for these thoughts, we have included more about the sediment sizes and different style river processes in the discussion. There was no vegetation present in the modelled inundated area, thus the vegetation could be excluded from the reasons affecting on these simulations.

RC: 9) Some sentences require sharpening / clearer expression (some are identified below). Check methods and results are consistently written in the past tense.

AC: Thank you for noting this. We have sharpened and clarified the expressions throughout the manuscript. We have also modified the language so that it is consistently in past tense in methods and results sections.

RC: 10) Data availability statement: since this is an open access journal I would expect the observed topography and model input files (as a minimum for the calibrated model runs for the small and moderate events) to be packaged up and available for download (perhaps with a dataset doi – see e.g. re3data.org to search suitable repositories). This would promote the "reproducibility" of the research and enable readers to re-run the models, if they were interested in analysing aspects of the results that are not presented in this paper.

AC: We agree, and it is possible for us to open the input topography, the input sediment grain sizes, the AOI of the calibration area, and the observed final topography of the calibration area. These all will be published in shapefile format so that it is easier for wider public to apply those, as not everyone might have Deflt3D model. Because some of the model input data files were from third parties, i.e. the discharge and the original laser scanned data, we are not able to open the whole model for the public.

RC: OTHER COMMENTS Title: Does changes need to be plural? "Gravelly" or "gravel bed" (also consider elsewhere) Delft2D: Delft3D is the name of the software but the simulations are executed in shallow water mode. Use "depth-averaged Delft3D" or similar phrasing rather than Delft2D.

AC: We have modified the title accordingly and also changed the word "gravelly" to "gravel bed" also elsewhere in the paper. The new title is:" Topographical change caused by moderate and small floods in a gravel bed ephemeral river - depth-averaged morphodynamic simulation approach". We have also modified throughout the paper that we are talking about the Delft3D model and its 2D implementation.

RC: P1 L15: Change performed to caused.

AC: This has been changed (P1 line 16).

RC: P1 L18: We pursue is an odd phrase in this context.

AC: While modifying the manuscript, we ended up deleting the sentence.

RC: P1 L20-22: This is the key research question – see my comment 1 above about emphasizing this throughout the manuscript.

AC: We have modified the aims based on your suggestion, and we are emphasizing this throughout the manuscript. Thus, the focus of the paper is now in analyses of the morphodynamics during the discharge events.

RC: P1 L28-31: Is there sufficient evidence from the two simulated events to extrapolate a conclusion about flood sequencing?

AC: Thank you for noting this. You might be right. We have deleted this conclusion.

RC: P2 L19: Yes, dynamics during high flows are difficult to measure but there are some attempts to do this e.g. Williams et al., 2015.

AC: The reference of Williams et al. 2015 is a study done in perennial rivers. Therefore we did not add the reference into this suggested sentence, but a little bit later in the next paragraph (P3 lines 19-21).

RC: P2 L23: Williams et al. citation should be 2016b not 2013

AC: This has been corrected thoughout the manuscript.

RC: P2 L28-29: How did the Hooke et al. (2005) model perform? Be more critical / analytical in the literature review examination (and this also goes for other parts of the introduction e.g. P2 L31 – why do the uncertainties arise?)

AC: We added description of how their model worked: "Their model worked well with simulations using moderately large discharges during clear water conditions, but discharge events with sediment loads had some tendency for excess deposition (Hooke et al., 2005)" (P3 lines 5-7). We also added related to the "P2 L31 comment" more explanation about the uncertainties and also about the quality of new measurement approaches in the introduction (P3 between lines 10-21). We have added more references and modified the background/introduction section into a more analytical form.

RC: P2 L32: References to the use of repeat surveys for morphodynamic modelling could be stated e.g. Lotsari et al., 2013; Williams et al., 2016a

AC: We have added the references (Lotsari et al., 2014a; Williams et al., 2016a) (P3 line 13).

RC: P3 L2: Broader references are needed here e.g. Milan et al., 2009

AC: We have added there now Milan et al., 2007; Vaaja et al., 2011; Calle et al., 2015; Kasvi et al., 2015; Kukko et al., 2015 (P3 lines 15-16).

RC: P3 L1: Within this paragraph you could emphasize more strongly that since ephemeral rivers can be surveyed whilst the river bed is dry, the topographic survey is characterized by lower errors because wet areas are usually associated with greater survey uncertainty.

AC: We have added in the introduction section following sentences to emphasize the advantages of the measurements in an ephemeral channel, and why there are less uncertainties than in perennial rivers. References have been also added (P3 lines 13-21): "Recently, the measurement techniques for deriving this calibration data for morphodynamic modelling have increased. One of them is accurate laser scanning (mobile and

terrestrial), which enables to capture the channel topography before and after flooding in detail (Milan et al., 2007; Vaaja et al., 2011; Calle et al., 2015; Kasvi et al., 2015; Kukko et al., 2015). The laser scanning enables rapid measurements of gravel bed rivers at the sub-grain level resolution (Milan et al., 2007). In ephemeral rivers the quality of topographical data can be very good and the uncertainties are less than in perennial rivers, either with laser scanning or traditional RTK-GPS measurements, because these rivers can be surveyed when the river bed is dry. The high uncertainties in topographical measurements of sub-water areas in gravel bed perennial river have been related particularly to the high bed load velocities and temporal variability of bed load (Williams et al., 2015)."

RC: P3 L15: I suggest emphasising the geomorphological research questions rather than the calibration exercise (see comment 1 above).

AC: We have refocused the paper and we now concentrate on the morphodynamics during the discharge events (see also our earlier responses).

RC: P4 L17: Where is the evidence that these sized grains were moved?

AC: We have clarified the sentence as follows (P4 line 32 onwards): "These two discharge events transported 12–41 mm sized gravel (D50 values) according to the measurements (see section 3.3 below), which had been performed in the areas of topographical changes. These have been also published at Calle et al. (2015). The movement of these gravels caused the development of the bar fronts (Calle et al., 2015)." Every sampling location had been evaluated in detail in Figs. 9 and 12 of Calle et al. (2015), where morphological mapping, DoD and simulation evidences are shown.

We had also already mentioned in methods section that "These measurements represented different active forms, from bars to the channel bed, which had evolved during the 2013 spring floods" (P9 line 33-P10 line 2).

RC: P5 L8: Are morphological and topographic both needed?

**ESurfD**
AC: You are right. We deleted the word "topographical".

RC: P5 L15: Clarify whether the grain size distributions were spatially distributed.

AC: The word "spatial" has been added to the sentence (P15 line 17).

RC: P6 L12: Clarify in the text whether the re-scaling with linear.

AC: We added a clarification "The hydrographs were re-scaled by using different multipliers to match the peak discharge calculated during the calibration procedure of the model (see Sect. 3.4 below, and from Calle et al., 2015)." (P6 lines 19-21)

RC: Figure 3 legend: More commentary is needed. Abbreviations need to be stated. A location map would be useful.

AC: We added more explanation of the abreviations to the figure caption. We also added the locations of the HWM measurements in the Fig.1.

The caption of the Fig. 3. is now as follows:" Figure 3: The hydrodynamic calibration results based on the coarse grid (1.51–5.31 m cells) and fine grid (0.76–3.03 m cells). In this figure, the high-water marks (HWM) and simulated water levels are presented. In addition, the trendlines fitted along these measurements and simulation results are shown. These calibration results of the fine grid were previously presented in Calle et al. (2015) by the same authors. The Qx1.3, Qx2 and Qx2.5 refer to the discharges scaled to match the peak discharge at the study site. Daily and 5 min refer to the observation interval of discharges at Vall d'Alba station. wl = water level based on HWMs and simulations, trendline = the trendline fitted in the measured HWMs and the simulation results. The location of these HWM measurement points have been shown in the Fig. 1."

RC: P8 L2: Sentence 2 – The three datasets need to be introduced before this sentence.

AC: The sentences have been clarified as "Three topography data sets were applied.

These were 1) initial topography (MLS, March 2012), 2) calibration topography between the floods (RTK-GPS, March 2013), 3) final validation topography after the floods (MLS, June 2013)." (P8 lines 10-12)

RC: P8 L18-19: Sentence not clear

AC: We have clarified the sentence as follows: "Cell sizes smaller than the "fine" resolution (i.e. 0.76–3.03 m) did not enhance the results, and those only increased the computational time. Thus, curvilinear grids of two resolutions, "coarse" 1.51–5.31 m cells and "fine" 0.76–3.03 m cells, were created from the topography measurement times." (P9 lines 4-7)

RC: P8 L25: Typo MSL

AC: This has been corrected as MLS (P9 line 9).

RC: P9 L16-20: Sentences not clear.

AC: We modified the sentences as follows: "These observed elevation and volumetric changes between the events were compared to the simulated changes. For calculating the volumetric changes the curvilinear grid topographies were needed to convert into regular grids. To minimize the errors, a 0.5 m regular grid's cell size was selected, as the original cells were mostly divisible by that value." (P9 lines 24-27)

RC: P9 L28: What guidance was used to sample a 10 cm layer?

AC: We used the criteria described in Bunte and Abt 2001, Page 188. After inspection of the surface layer in a near pit, and considering the larges particles, we estimated an embedded depth of 10 cm. We have added clarification in the manuscript (P10 lines 4-5): "The upper layer-sublayer contact was established following the criteria described by Bunte and Abt (2001) considering the size of the largest particles embedded depth."

RC: P9 L30: Is this difference for the upper or sub layer?

AC: Those were all upper layer samples (P10 lines 5-6).

RC: P10: How many size fractions were used in the model?

AC: D50 values were used in the model. We added a sentence: "The D50 grain size values were used in the model." (P11 line 1)

RC: P10 L11: It would be useful to refer to a map here.

AC: We are referring to a map of Calle et al. 2015, and we have now included the sample locations in our Fig. 1 too (P10 line 11).

RC: P11 L1: Insert "solving" after "for".

AC: During the shortening of the methods section, this sentence was deleted.

RC: P11 L5: This event has previously been referred to as March 2013 rather than 6.3.2013. Be consistent

AC: This has been modified as "...WL trendline, of the 6th of March 2013 discharge peak situation." (P11 line 13). We have also checked the consistency throughout the manuscript.

RC: P12 L12: Quantify "sufficiently well"

AC: This was vague selection of words from us. We modified the sentence as "These simulations also reproduced the water levels" (P12 line 19).

RC: Table 6: Separate volumes and

AC: We have separated "the volumes" and "the difference compared to observed" into their own columns. We have splitted this table into two. The original whole table can be seen from the supplementary data of the paper (there the Table 3), and the data related to observations and the simulations 9–10 is also presented in the new Table 5 in the results section of the main manuscript.

RC: Fig 4 / 5: Be consistent in use of "After" in figures (top left corner) but not in other sub-figures.

AC: We have modified the former Figs. 4 and 5 by deleting the word "after". We have moved these two figures in the supplementary material, where they are Figs. 1 and 2.

RC: P20 top paragraph: The morphological change description here is interesting and I think the results / discussion would benefit from more analysis of this style.

AC: We have refocused the paper and thus also modified the results to include/concentrate more on morphological changes during the discharge events. We have also modified the discussion, conclusions and aims.

RC: Section 5.1: This discussion needs to be more closely integrated into the discussion in other, similar Delft3D (or other graded sediment morphological model) calibration findings.

AC: We have morified the 5.1 section thoroughly. We have added more references and discussion related to other studies done with Delft3D model, and the calibration of Delft3D/graded sediment morphological models in other studies (see section 5.1).

RC: P25 L11: "Unique" – from sample of 2?

AC: We have modified the sentences as (P25 line 1): "Our study has shown that geomorphic responses to the two analysed discharge events differed."

RC: P25 L31: "differs" – explain why

AC: We had explained in the paragraph, that they observed most erosion during the rising and peak flow phases, as in Rambla de la Viuda the receding phase was found also important for the channel changes (P25 lines 12-21). We have now also added the following sentences in this paragraph to explain this in more detail: "They found only some scour during sustained high flows following the flood peak (Gendaszek et al., 2013). Noteworthy is that they applied one sensor per reach, and thus the site selection could have greatly affected on their results."

RC: REFERENCES Lotsari, E., D. Wainwright, G. D. Corner, P. Alho, and J. Käyhkö
(2013), Surveyed and modelled one-year morphodynamics in the braided lower Tana River, Hydrological Processes, 28(4), 2685-2716. doi: 10.1002/hyp.9750. Milan, D. J., G. L. Heritage, and D. Hetherington (2007), Application of a 3D laser scanner in the assessment of erosion and deposition volumes and channel change in a proglacial river, Earth Surface Processes and Landforms, 32(11), 1657-1674. doi:10.1002/esp.1592. Stecca, G., R. Measures, and D. M. Hicks (2017), A framework for the analysis of noncohesive bank erosion algorithms in morphodynamic modeling, Water Resources Research, doi: 10.1002/2017WR020756. Schuurman, F., W. A. Marra, and M. G. Kleinhans (2013), Physics-based modeling of large braided sand-bed rivers: bar pattern formation, dynamics and sensitivity, Journal of Geophysical Research: Earth Surface, 118(4), 2509-2527. doi: 10.1002/2013jf002896. Wheaton, J. M., J. Brasington, S. E. Darby, and D. A. Sear (2010), Accounting for uncertainty in DEMs from repeat topographic surveys: improved sediment budgets, Earth Surface Processes and Landforms, 35(2), 136-156. doi: 10.1002/esp.1886. Wheaton, J. M., J. Brasington, S. E. Darby, A. Kasprak, D. Sear, and D. Vericat (2013), Morphodynamic signatures of braiding mechanisms as expressed through change in sediment storage in a gravel-bed river, Journal of Geophysical Research: Earth Surface, 118(2), 759-779. doi: 10.1002/jgrf.20060. Williams, R. D., C. R. Rennie, J. Brasington, D. M. Hicks, and D. Vericat (2015), Withinevent spatially distributed bed material transport: linking apparent bedload velocity to morphological change, Journal of Geophysical Research: Earth Surface, 120(3), 604-622. doi: 10.1002/2014JF003346. Williams, R. D., J. Brasington, and D. M. Hicks (2016a), Numerical Modelling of Braided River Morphodynamics: Review and Future Challenges, Geography Compass, 10(3), 102-127. doi: 10.1111/gec3.12260 Williams, R. D., R. Measures, D. M. Hicks, and J. Brasington (2016b), Assessment of a numerical model to reproduce event-scale erosion and deposition distributions in a braided river, Water Resources Research, 52(8), 6621-6642. doi:10.1002/2015WR018491.

AC: We have gone through these references and added most of these (i.e. the most suitable ones) into the manuscript and its reference list.
* * *
REFEREE #2:

RC: The paper provides a detailed account of the calibration of a numerical model (a 2d implementation of Delft 3D) in order to simulate the geomorphic changes during flash flood events. The main novelty and significant findings of the work are the predictions of when sediment transport occurs during flood events of different magnitudes, based on a model simulation calibrated versus observed changes. It is an interesting and thorough piece of research. However, the purpose of the work and the findings and conclusions get lost in a highly detailed description of the calibration method. The calibration itself is not particularly novel, although as it includes observations spanning two events it is more robust than many other examples. Much of this detail could be moved to supplementary materials, and a concise summary provided in the main manuscript. This would allow the manuscript to be focussed more on the simulations themselves, what they show, and why this is important (I felt this last point was not made clearly enough throughout). The authors at several points refer to sensitivity testing performed on the model. Although evidently there were some tests performed to assess model behaviour differences with some variations, these are not sufficiently thorough to be considered a sensitivity analysis, and falls short of the level of analysis performed by operators in other modelling fields. For example, the tests performed would not ascertain any non-linear interactions between the parameters being tested. For interest, Ziliani et al (2013) provides a useful and efficient methodology for screening model parameter sensitivity in reach-scale geomorphic models, which might be useful for future work. The authors should be careful using terms such as sensitivity analyses, and the conclusions they draw from the tests should not be presented with as much certainty as they presently are. The tests may be useful for informing the calibration process, but cannot be relied upon to suggest anything concrete about the model behaviour.

AC: Thank you for these thorough comments. We have refocused the manuscript to emphasize the geomorphological findings during the discharge events (also referee #1

had suggested that). Thus, we have modified the aims, results, discussion and conclusions accordingly. We have moved the morphodynamic model's calibration results into the supplementary material and are now presenting a summary of the calibration under the methods section (3.5). Thus, we have also moved the former Figs. 4 and 5 and Tables 5 and 6 into this supplementary material (now Figs. 1 and 2, and Tables 2 and 3 in the supplementary material). We added also Table 4 into the supplementary material (there it is the Table 1), all thought it is also presented in the main manuscript as Table 4. We also present the former Fig. 6 in the supplementary material (Fig. 3 there) and parts of it in the manuscript (now Fig. 4 there), as these are needed in both of the documents so that it is possible to read those separate documents independently. We have also changed the wordings throughout the paper, i.e. we are not anymore talking about the sensitivity analyses, only about calibration. We have now also emphasized more in the introduction section, why this research is important. We have also modified the conclusions so that the emphasis is on geomorphological findings and deleted conclusions related to model calibration results.

RC: Below are some notes for correction or additional clarification: Page 1, Line 15 – use "caused by the flood" or "resulting from the flood" instead of "performed by"

AC: We have modified it as "caused by a flood" (P1 line 16).

RC: Page 1, Line 16 (and throughout) – strictly speaking, a 2D implementation of Delft 3D

AC: We have modified this throughout the manuscript. Thank you for noticing this mistake of ours.

RC: Page 1, Line 24 – As explained above, I don't think a rigorous enough sensitivity test has been performed to make these conclusions with such certainty

AC: Thank you for pointing this out. We agree, and we have modified the manuscript so that the sensitivity analyses are not anymore mentioned, and we talk only about

calibration. Also the calibration section has been reduced: a summary of the morphodynamic model's calibration is now presented in the methods section, and most of the material is presented in as the supplementary material for this paper. We have also modified the abstract based on the modifications done to the manuscript and its conclusions.

RC: Page 1, Line 24-25 – Is the total load equation probabilistic? Or is it implemented in a deterministic way too?

AC: You are right the total load equation had been implemented in a deterministic way. We have removed the sections mentioning the deterministic way. We have modified this throughout the manuscript.

RC: Page 2, Line 13-16 – I'd like to see something here to explain why this is important. Why does it matter that we know this?

AC: We have added the following explanations about the importance of the study in the introduction (P2 lines 3-8): "The costs of these floods due to their catastrophic nature include major economic, social and environmental aspects (Petersen, 2001). These are caused by both hydro- and morphodynamics during the discharge events. For reducing the emergency costs and enhancing the preventive flood mitigation measures, understanding of the forces of the flow and related channel changes throughout the discharge events would be important. For being able to allocate the measures temporally most efficiently, the understanding of the timing of the morphodynamics and their effects to the adjacent river environment throughout the discharge events are needed."

AND

P2 lines 20-25: "Case studies have reported most erosion during the rising and the peak flow phases in perennial rivers (e.g. Gendaszek et al. 2013), but similar knowledge for ephemeral river channels is still low. It would be important to understand the capacities of ephemeral rivers for sediment deposition and flooding due to the combined effects of water flow and sediment transport during flood situations. If the erosion and deposition would be in total greater during the receding phase than in rising phase, the receding phase should not be ignored while planning flood mitigation measures."

RC: Page 2, Line 30-34 – The authors are also in danger of overparamaterising the model by using a single calibration against a small set of observations. It risks equifinality with the model matching the data well, but there are many ways that the changes could have come about.

AC: We added "of different consecutive flood situations" (P3 lines 9-10), so that it is here pointed out that the calibration and validation can be done by using consecutive flood situations.

RC: Page 3, Line 19 – With two events the authors could have calibrated with one and validated with the other as an extra check.

AC: This is what was done, and we have reworded the text. We modified the sentence as "Thus, the model is calibrated with data from moderate magnitude event, and then validated based on the consecutive low-magnitude event." (P4 lines 5-6).

RC: Page 4, Figure 1 – It would be more useful to see the reach in context with the wider catchment here than its rough location in Spain. How big is the catchment and where is the reach in relation to the gauging station used?

AC: The watershed boundaries, observation station location, detailed study site location, Maria Christina reservoir and river network have been included in the new Fig. 1. The total water shed size and the increase in the watershed size between the gauging station and the study site had been already mentioned in the text.

RC: Page 6, Lines 10-16 – What is happening between in 18km between the gauge and the reach to increase the discharge. Are there more channels flowing in? Has there been more rainfall in between? Convective events are likely not to span 18km so rainfall will not be uniform. This is key as the authors make the assumption that

the hydrograph is the same shape when flow makes it to the reach, just scaled larger, yet the reasons for the increased discharge will also likely change the shape of the hydrograph and this potentially invalidates the calibration and the conclusions. More detail needs to provided here to justify the above assumption.

AC: The rain producing floods in this catchment are not typical summer convective events, but Mesoscale Convective Complexes (autumn heavy rains), and those may be up to 100 km in diameter. The rains producing these floods are autumn and spring, and are related to Atlantic fronts, which do not give much spatial variability. We have added a more detailed description about the rain events as follows (P4 lines 13-15): "The rain producing floods in this catchment are caused by mesoscale convective complexes (autumn heavy rains), and those may be up to 100 km in diameter. The rains producing these floods occur in autumn and spring, and are related to Atlantic fronts, which do not give much spatial variability."

AND

P4 lines 26-32: "The two events under study occurred in spring 2013. These were caused by two rain events, which were recorded with 5 minute interval at the precipitation and gauging station provided by the Automatic System of Hydrologic Information (SAIH-Jucar) at Vall d'Alba (Fig. 1 and Calle et al., 2015). The first rain event (March 2013) started on 4th of March lasting for three days with a total 70 mm. The second rain event (April-May 2013) started on 27th April and lasted four days with a total of 72 mm. These rain events caused flows that lasted 13 days (started at 12:00 on 5th March) and 9 days (started 14:10 on 30th April), respectively. The peak discharges of 23 (at 11:05 on 6th March) and 12.5 m3 s-1 (at 9:05 on 1st of May) were registered in the gauging station of Vall D'Alba, respectively."

We can verify that the hydrograph shape at the study site of Rambla de la Viuda is similar to the hydrograph at Vall d'Alba station (see the supplementary material entitled "hydrograph shape"). We had installed water level sensors at the study site later in

year 2014. Based on these we can verify that the hydrograph shape of the study site of Rambla de la Viuda is similar to Vall d'Alba observation station (see the locations from the Fig. 1 of the main paper document) and that the water level is much higher at the study site than the Vall d'Alba station. Thus, these measurements prove that the discharge has to be also much higher in the study area than at Vall d'Alba. In addition, based on the roughness calculations with Limerinos equations and the model calibration results, the Qx2 discharge was the correct one to be applied in the model. Based on the Limerinos equation calculations, the Qx2 was expected to work the best due to the average nature of the derived roughnesses and inclusion of bedform and bank roughness effects.

In addition, it was possible to calculate the daily discharge curve for María Cristina reservoir, which locates downstream of the study site of ours (see the supplementary material). When comparing the March 2013 discharge peak of Qx2 discharges (Fig. 5 in the main manuscript), the resemblance to the daily discharges of María Cristina reservoir can be seen. Due to the fact that only daily data was available from Maria Christina reservoir, the curve of the reservoir is flatter than the one at the study site. Also calibration curves for the reservoir and water loss by infiltration are unclear. Despite these, we can be confident that the shape of the hydrogprah at the study site was realistic and that the Qx2 discharge would be the correct discharge to apply at Rambla de la Viuda study site.

In the main manuscript, we also refer to this supplementary material regarding the hydrograph shape.

RC: Page 6, Lines 17-22 – By calibrating the discharge against water levels there is another assumption that the bed levels were static and had no influence on the water levels in the flood (despite in the discussion stating that bed changes are an important influencer on water levels). This will lead to some uncertainty cascading to the simulations.

AC: This had been taken into account, and the water level was also checked during the morphodynamic simulations. We modified now the sentence in section 3.5 (P13 lines 20-21): "These 61 simulations included simulations with Qx1.3, Qx2 and Qx2.5 discharge hydrographs. During these simulations the match of the simulated water levels to HWMs was also checked. The roughness values were still valid." As the water level had been discussed in earlier section, we had decided not to show these in a figure anymore here, as this section concentrates on topographical comparisons. Also because other referees' comments favored the shortening of the calibration methods section, we decided at this stage not to add any further figures in the methods section.

RC: Page 9, Line 30-31 – What is the expected level of uncertainty in the measurements – is 1.6mm difference significant?

AC: We were not able to detect the level of uncertainty of grain size measurements. In 2012, the measurements were done with Wolman sampling, and in 2014 with sieving. Thus, only the actual difference was possible to show. The Wolman sampling method is more uncertain than the sieving.

RC: Page 11, Line 15-16 – If the Manning's n values were set according the geomorphological elements, did they alter with changes during the model runs, or were they stationary?

AC: Those were stationary. A clarifying sentence has been added in P11 line 25 – P12 line 1: "Note that the assignment of the n values was static throughout the simulations, i.e. the spatial distribution of the roughness values was not possible to change between simulation time steps."

RC: Page 11 Line 18-20 and Table3 – either present the values in the table in the order of Qx1.3, Qx2 and Qx2.5, or present the values in the text in the order of the table.

AC: The order of the values in the Table 3 has been changed.

RC: Page 12, Line 31-32 – Tells you something about model sensitivity to parameters,

but not a great deal.

AC: The sentence has been decided to delete, as it did not bring any additional information when compared to the rest of the text. We also overall reduced the text in methods and calibration section of the modphodynamic model.

RC: Page 13, Line 1 – Need to know what the criteria was for this! "best and most interesting" not a rigorous methodology. The authors might have rejected useful information by arbitrarily labelling it as not interesting.

AC: You are right, this was vague. The sentence has been deleted during the modification of the manuscript. We now describe in P14 lines 17-21 "... 11 simulations done during calibration were selected to be presented in the supplementary material of this paper (see also Table 4). Reason for their selection was that these 11 simulations showed the effects of grain size (before [2012] and after [2014] floods, and grain sizes from different layers [2014]), grid size (coarse: 1.51–5.31 m, fine: 0.76–3.03 m), transverse slope (user defined coefficients in the bed load transport equations: default 1.5 and increased to 3) and transportation equations (Engelund-Hansen [EH], Meyer-Peter and Müller [MPM]) on model performance (Table 4, and supplementary material)."

In the supplementary material, these sentences are now: "The best simulation results (from the simulations 1, 2, 3 and 10) in relation to the surveyed volumetric changes were achieved with the fine grid simulation (number 10)".

RC: Page 13, Line 2 – Not a sensitivity analysis.

AC: We have modified this as morphodynamic model's calibration.

RC: Figures 4, 5 and 6 – These would be easier to interpret if they showed the changes over the whole reach, not just in the area used for the analysis. This could still be identified as in Figures 1 and 8.

AC: We had calibrated the model based on this area, where we had data from all the topography measurement times (2012, March 2013 and June 2013). Thus, the reason

for showing the calibration area in these figures is, because we were able to perform the goodness-of-fit only in this area. We have modified the manuscript so that we are not anymore talking about the "analyzed area" but about "calibration area". We have now added the whole simulation area to in the new Figs. 6 and 7, i.e. in the results section related to the morphodynamic analyses. In these two figures, the results of the best simulation are shown. We are now showing the whole simulation area also in Fig. 1.

RC: Page 19, Line 18 – "satisfactory" not "satisfying"

AC: During the shortening of the methods section, the sentence had been deleted from the main manuscript.

We have modified this accordingly in the text of supplementary material, where this sentence had been moved.

RC: Page 19, Line 25 – or other factors had an influence? Interaction with other parameters?

AC: We tried the simulations with many different parameters, and the MPM did not produce much movement with any of the parameterizations. We decided to just show in the manuscript (now in the supplementary material) this as an example of the unsuccessful simulations with the MPM equation. The MPM simulation results have been moved to the supplementary material.

RC: Page 22, Line 8 – Has it been shown that it is reliable? It's shown it is able to reproduce the events it was calibrated against, but not others.

AC: We modified the sentence as "The reliability of the model, which is calibrated against the events under interest, can be improved with the quality and temporal density of the available calibration topography, i.e. pre- and post-flood bedform geometries." (P21 lines 25-27).

RC: Page 23, Line 2 – Yes, validation of the calibrated model versus independent

events is needed to claim the model is reliable.

AC: Yes, we agree. We have modified the discussion, and modified the sentence and paragraph as follows (P21 line31-P22 line 6) "However, despite the high quality data from two events at Rambla de la Viuda, we think that further research with multiple yet-to-come events needs to be run to assess the repeatability and validation of the model even better. For example, at Rambla de la Viuda, large floods have not yet occurred since the beginning of the MLS measurement approaches. As also earlier has been stated (Verhaar et al. 2008; Lotsari et al., 2015), the roughenss conditions defined for small discharge events, might not be suitable for simulating extreme events. Therefore, the work and refinement of the model will continue, and the applicability of the model for larger floods will be tested, when validation data will be available."

RC: Page 23, Line 4 – Yes, the uncertainties need to acknowledged, maybe a little more prominently then there are at present.

AC: We have addressed the uncertainties in more detail throughout the discussion section (5.1), and we have for example added the following text in the discussion (P22 lines 9-27): "The uncertainties of the present model approach thus relate to the lack of sediment transport, flow and topographical data during the events. However, the selected initial boundary conditions seem to be congruent with the flooding mechanisms of Rambla de la Viuda and also with other published works, i.e. Williams et al (2016b), for loosely consolidated sand and gravel. So, the modelled flow carried each sand sediment fractions (suspended) adapted to the local flow conditions at inflow boundary, and the model assumed that very little accretion or erosion was experienced near the model boundaries. Based on measurements, the presence of sand size particles and their concentration in the study site were also almost non-existent, and no channel changes occurred at downstream boundary of the simulation area. Thus, this equilibrium load condition was considered valid, and it was also the only option for the present modelling approach, as no input suspended load measurements were available. Similarly as Williams et al. (2016b) state, the model results, i.e. the modelled deposition

and erosion, when compared to observations, could have been possibly enhanced if the input suspended sediment load observations would have been available. However, according to Sanyal (2017) the sediment transport is always inherently approximate in nature, and sediment load added to the model causes uncertainties to the results, despite detailed sediment load measurements have been used as model input.

Despite this lack of the data during the flood events, we had a good control on sediment volume and gravel particle-size moving downstream as the forefront lobe prograded over a flat valley bottom (as gravel bed had been mined). Total volume input and total transport rates observed in earlier study of Rambla de la Viuda by Calle et al. (2015), had already proved the high availability of sorted gravel particles, and were the basis of the decisions made while building up the model. As already mentioned, the channel changes at the output downstream boundary of the studied reach was zero."

RC: Page 23, Line 10 – The authors have not provided enough detail to evidence that the hydrograph is of "a known shape". It may well be, but the gauge is 18km away with half the discharge (according to the calibrations).

AC: Based on the later (2014) measurements with water level sensors at the study site and the nature of the rain events, we can state that the hydrograph shape was similar to Vall d'Alba station (see above our earlier more detailed responses, and the supplementary material, regarding the hydrograph shape).

RC: Page 24, Line 13 – Both equations are implemented determinisitically are they not?

AC: You are right, they are implemented deterministically. We have removed the sections mentioning the deterministic way. We have modified this throughout the manuscript.

RC: Page 25, Line 13 – Are the parameter sets non-stationary or is it the initial conditions which need to be set for each event?

AC: We hope that we understood your comment correctly, and we modified the sentence as "Previously this uniqueness of geomorphic response has been shown for perennial rivers (Pitlick, 1993), where also topographical and sedimentary data has been applied as initial conditions." (P25 lines 2-4)

RC: Page 27, Line 18 – Instead of "goodness" use "feasibility"

AC: We ended up modifying and shortening the conclusions section, and this sentence was deleted during the modifications.

RC: Page 28, Line 6 – Both implemented deterministically?

AC: You are right, they are implemented deterministically. We have removed the sections mentioning the deterministic way. We have modified this throughout the manuscript.

RC: Page 28, Line 8 – This is a key point, and should be made more prominent earlier in the manuscript as part of the reason why this research is important.

AC: We have refocused the paper and increased the morphological analyses throughout the paper (see also our previous responses). We have raised these as key points already in the introduction section.

RC: Page 28, Line 16-18 – Can this be said with such certainty, or does it just apply to this reach for these events (and also with these parameters) Ziliani et al (2013) - doi:10.1002/jgrf.20154

AC: We added the following sentences: "These differences between the events were at least evident with these two events and the applied parameterization. However, further studies of this possible hysteresis effect are needed from multiple discharge events." (P28 lines 9-10).

RC: Thank you for an interesting study and manuscript, I look forward to seeing the revised version.

AC: Thank you for your thorough comments, those were really helpful and enhanced the paper greatly.
* * *
REFEREE #3

RC: Overview This paper uses a 2D morphodynamic model to assess the impact of small and moderate floods on the evolution of ephemeral rivers. This is an interesting topic which will be of use to the wider community however in its present form I do not think it is suitable for publication. The authors spend over half of the paper describing the model set up and calibration and do not really address the original question. Given a lot of the model inputs were from a previously published paper a lot of the rather dense description could be cut from the paper to allow more time for a detailed analysis of the impacts of flood characteristics. There is also little time devoted to discussing the applicability of this model to scenarios other than the very detailed description in the paper for which there is good input boundary data. Detailed comments and queries are below which the authors needs to address if this paper is to be published.

AC: Thank you for the thorough comments. We have modified the manuscript thoroughly and refocused the paper in introduction (also aims), results, discussion and conclusions sections to emphasize the geomorphological findings (see also responses to referee #1 and #2 comments). The paper now aims at analyzing (P3 line 30 onwards) "the evolution of a gravel bed ephemeral river channel (Rambla de la Viuda, Spain) during consecutive, moderate- (March 2013) and low-magnitude (May 2013), discharge events, by applying a morphodynamic modelling (Deflt 3D) approach. Based on the simulations, we analyze 1) the timing of river bed erosion and deposition in relation to the flow hydrograph phases during moderate- and low-magnitude discharge events, 2) the hydraulic characteristics (e.g. shear stress) explaining these channel and bed-form morphodynamics, and 3) the prevailing fluvial processes, and related sediment transport routing, during these different magnitude discharge events in a gravel bed

ephemeral stream. Understanding of these processes would be needed particularly for river management works and flood mitigation purposes."

Along with these changes, we have modified the results section so that we have moved the morphodynamic model's calibration results to the supplementary material (as the second reviewer also suggested), and present only their summary in the methods section. We have added more results related to the morphodynamics (channel bed elevation changes and bed shear stresses) to the manuscript so that we were able to refocus the paper.

We find it important to include the description of the data and hydrodynamic model in the methods of the main manuscript, as otherwise, the manuscript would not be readable independently, i.e. without Calle et al. (2015) paper. In addition, Calle et al. (2015) does not have all the required data explanations. We have referred to Calle et al. (2015) in every case, when that is possible, such as related to the details of the topographical measurements and error analyses. We have modified the discussion section and added there more discussion about the applicability of the model to other (larger) flood scenarios.

RC: Page 2 Line 4 – I would query the word greatest and authors should consider an alternative

AC: We have modified the word as "large" (P2 line 10).

RC: Page 2 – Line 19-21 – three papers might be worth reviewing – although not on the context of ephemeral rivers they give useful context Viparelli et al (2011) 'A model to predict the evolution of a gravel bed river under an imposed cyclic hydrograph and its application to the Trinity River' WRR An et al (2017) 'Gravel-bed river evolution in earthquake-prone regions subject to cycled hydrographs and repeated sediment pulses' ESPL An et al (2017) 'Effect of grain sorting on gravel bed river evolution subject to cycled hydrographs: Bed load sheets and breakdown of the hydrograph boundary layer' JGR ES

AC: Because the papers do not relate to the ephemeral rivers, in our opinion, they fitted better to discussion section. Therefore, we have added these into the discussion section (P23 lines 1-4), instead of the introduction.

RC: Page 2 Line 30 – when you say between and after flood topographies do you mean pre and post flood topographies?

AC: Yes, we mean pre- and post-flood topographies. We have modified them as "pre- and post-flood topographies" on P3 line 9.

RC: Page 3 Line 12 – use alternative phrasing for high/ large floods

AC: During the modification of the paper, we have deleted the whole sentence.

RC: Page 3 Line 12/13 – the sentence beginning in addition does not make grammatical sense

AC: During the modification of the paper, we have deleted the whole sentence.

RC: Page 3 Line 25/ 26 – consider 'The river has a braided pattern associated with a high sediment supply' instead of current wording

AC: We have modified this accordingly (P4 lines 9-10): "The river has braided pattern associated with a high sediment supply (Calle et al., 2015)."

RC: Page 4 Line 15 – how far away was the gauging station from the study site?

AC: The detailed gauging station and study site locations have been added into the new version of the Fig. 1.

RC: Page 4 Line 15 – you say the discharges at the field site were estimated to be higher but how much? How did you estimate this?

AC: During the modification of the paper (also based on comments of referees #1 and #2) the sentences were removed. We have now describe the discharge and rain events more in detail in this paragraph (P4 lines 26-32): "The two events under study occurred

in spring 2013. These were caused by two rain events, which were recorded with 5 minute interval at the precipitation and gauging station provided by the Automatic System of Hydrologic Information (SAIH-Jucar) at Vall d'Alba (Fig. 1 and Calle et al., 2015). The first rain event (March 2013) started on 4th of March lasting for three days with a total 70 mm. The second rain event (April-May 2013) started on 27th April and lasted four days with a total of 72 mm. These rain events caused flows that lasted 13 days (started at 12:00 on 5th March) and 9 days (started 14:10 on 30th April), respectively. The peak discharges of 23 (at 11:05 on 6th March) and 12.5 m3 s-1 (at 9:05 on 1st of May) were registered in the gauging station of Vall D'Alba, respectively."

We have added clarification related to the discharge hydrographs in the methods section (P6 line 14-21): "At the study reach, the hydrograph peak discharge was estimated from continuous lines of flotsam (i.e. high-water marks, HWM) emplaced by the floodwater. The hydrograph shape of the study site can be assumed similar to Vall d'Alba gauge station, due to the widespread continuous character of the rain events. The hydrograph shape was verified to be similar to Vall d'Alba observation station also by recent installation of water level sensors (in late 2014) in the study site (see supplementary material). The HWM left by the March 2013 discharge event provided evidence that the peak discharge was greater at the study site than that measured at Vall d'Alba (Figs. 1 and 3). The hydrographs were re-scaled by using different multipliers to match the peak discharge calculated during the calibration procedure of the model (see Sect. 3.4 below, and from Calle et al., 2015)."

AND

Sentences starting P12 lines 8-11 "The "Qx2" (observations multiplied by 2) water surface elevation matched the HWMs using an average n-values calculated from Limerinos (1970) equations. Also, the effects of bedform (from +0.01 to +0.015) and bank roughness (+0.02) had been added to these average n-values (Chow, 1959; Acrement and Schneider, 1989)", P12 lines 14-15 "Based on the Limerinos equation calculations, the Qx2 was expected to work the best due to its average nature and inclusion

of bedform and bank roughness effects."

Also, in Calle et al. (2015) (where the same hydrodynamic model had been used), the Qx2 was proven to work the best regarding the velocities and shear stresses needed for sediment movement. In Calle et al. (2015) paper we demonstrated with geomorphological indicators, i.e. grain size and observed morphologic change, that our model calibration was correct. See Fig. 12 in Calle et al. (2015), where this is discussed in terms of velocities. Thus, the same parameters worked also for the subsequent morphodynamic simulation that resulted in this paper.

RC: Page 6 Lines 9-15 - you have assumed that the discharge between the gauging station and the reach is increasing but that the hydrograph shape remains the same but how is this so? This has important implications for the validity of the calibration of your model. Much more detail is needed to justify this assumption

AC: We can verify that the hydrograph shape at the study site of Rambla de la Viuda is similar to the hydrograph at Vall d'Alba station (see the previous responses and the supplementary material entitled "hydrograph shape"). We had installed water level sensors at the study site later in year 2014. Based on these we can verify that the hydrograph shape of the study site of Rambla de la Viuda is similar to Vall d'Alba observation station (see the locations from the Fig. 1 of the main paper document) and that the water level is much higher at the study site than the Vall d'Alba station. Thus, these measurements prove that the discharge has to be also much higher in the study area than at Vall d'Alba. In addition, based on the roughness calculations with Limerinos equations and the model calibration results, the Qx2 discharge was the correct one to be applied in the model. Based on the Limerinos equation calculations, the Qx2 was expected to work the best due to the average nature of the derived roughnesses and inclusion of bedform and bank roughness effects.

In addition, it was possible to calculate the daily discharge curve for María Cristina reservoir, which locates downstream of the study site of ours (see the supplementary
material). When comparing the March 2013 discharge peak of Qx2 discharges (Fig. 5 in the main manuscript), the resemblance to the daily discharges of María Cristina reservoir can be seen. Due to the fact that only daily data was available from Maria Christina reservoir, the curve of the reservoir is flatter than the one at the study site. Also calibration curves for the reservoir and water loss by infiltration are unclear. Despite these, we can be confident that the shape of the hydrgprah at the study site was realistic and that the Qx2 discharge would be the correct discharge to apply at Rambla de la Viuda study site.

We have added clarification related to the discharge hydrographs in the methods section (P6 line 14-21): "At the study reach, the hydrograph peak discharge was estimated from continuous lines of flotsam (i.e. high-water marks, HWM) emplaced by the floodwater. The hydrograph shape of the study site can be assumed similar to Vall d'Alba gauge station, due to the widespread continuous character of the rain events. The hydrograph shape was verified to be similar to Vall d'Alba observation station also by recent installation of water level sensors (in late 2014) in the study site (see supplementary material). The HWM left by the March 2013 discharge event provided evidence that the peak discharge was greater at the study site than that measured at Vall d'Alba (Figs. 1 and 3). The hydrographs were re-scaled by using different multipliers to match the peak discharge calculated during the calibration procedure of the model (see Sect. 3.4 below, and from Calle et al., 2015)."

RC: Page 6 Lines 17-22 – you have assumed that bed level has not changed when you have calibrated discharge to water levels – how valid is this assumption?

AC: The water levels, which were based on the HWM measurements can be assumed valid, as the bed elevation did not change at the area, where the downstream water level boundary was defined (see also P11 lines 18-22): "The Limerinos equation was applied for the whole range of water levels (i.e. hydraulic radiuses) for the March 2012 and March 2013 grain sizes of D50 and D84. These represented the best of the preceding conditions of the March 2013 and May 2013 flows. The Limerinos calculations

were performed for a cross-section located at the downstream side of the simulation area. This cross-section was defined from 2012 geometry, i.e. pre-flood geometry, which remained unchanged during the 2013 discharge events."

RC: Figure 3- it is unclear how this relates – more detailed description needed

AC: We have clarified the figure caption: "Figure 3: The hydrodynamic calibration results based on the coarse grid (1.51–5.31 m cells) and fine grid (0.76–3.03 m cells). In this figure, the high-water marks (HWM) and simulated water levels are presented. In addition, the trendlines fitted along these measurements and simulation results are shown. These calibration results of the fine grid were previously presented in Calle et al. (2015) by the same authors. The Qx1.3, Qx2 and Qx2.5 refer to the discharges scaled to match the peak discharge at the study site. Daily and 5 min refer to the observation interval of discharges at Vall d'Alba station. wl = water level based on HWMs and simulations, trendline = the trendline fitted in the measured HWMs and the simulation results. The location of these HWM measurement points have been shown in the Fig. 1."

RC: Page 8 Lines 15 – 21 better justification is needed of cell sizes – e.eg what do you mean 'did not make more difference to the results'?

AC: The sentence has been modified as follows (P 9 lines 4-7): "Cell sizes smaller than the "fine" resolution (i.e. 0.76–3.03 m) did not enhance the results, and those only increased the computational time. Thus, curvilinear grids of two resolutions, "coarse" 1.51–5.31 m cells and "fine" 0.76–3.03 m cells, were created from the topography measurement times."

RC: Page 8 Line 25 – MSL or MLS?

AC: We have corrected this as MLS (P9 line 9).

RC: Page 9 – if the water level did not reach the high bank elevations why add that 2009 DEM results to the model – what does it add?

AC: This was done so that the high banks would be as realistic as possible. The model was built up for not only to calibrate these moderate and low magnitude events but also for being able to later use the same model to simulate bigger flood events, which have not yet occurred since the beginning of our observations in the area. Thus, measurements and observations are still ongoing at Rambla de la Viuda.

RC: Page 9 Line 16 -17 – what do you mean ' the capabilities of the model to result correct channel bed elevations'? This sentence needs restructuring.

AC: As we shortened the methods sections, we also ended up deleting this paragraph.

RC: Page 9 lines 16- 20 – the meaning of this section of text is unclear

AC: As we shortened the methods sections, we also ended up deleting this paragraph.

RC: Page 9 Line 30 – is this the difference between the armour and sub surface layer? What was the difference in the D84?

AC: These armour and sub-layer values were not possible to define between 2012 and 2014, as there was no sub-layer measurements in 2012. The D84 values of 2014 have been added to Table 2.

RC: Page 13 – what do you mean 'best and most interesting results'? This surely needs justification? What do you deem best or most interesting?

AC: You are right, this was vague. The sentence has been deleted during the modification of the manuscript. We now describe in P14 lines 17-21 ". . . 11 simulations done during calibration were selected to be presented in the supplementary material of this paper (see also Table 4). Reason for their selection was that these 11 simulations showed the effects of grain size (before [2012] and after [2014] floods, and grain sizes from different layers [2014]), grid size (coarse: 1.51–5.31 m, fine: 0.76–3.03 m), transverse slope (user defined coefficients in the bed load transport equations: default 1.5 and increased to 3) and transportation equations (Engelund-Hansen [EH], Meyer-Peter and Müller [MPM]) on model performance (Table 4, and supplementary material)."

In the supplementary material, these sentences are now: "The best simulation results (from the simulations 1, 2, 3 and 10) in relation to the surveyed volumetric changes were achieved with the fine grid simulation (number 10)".

RC: Page 13 Lines 2-3 define better performance?

AC: This has been modified as (P13 lines 25-26): "Due to the better correspondence of simulations with Qx2 discharges to the observed channel evolution, all these selected morphodynamic simulations had these input data."

RC: Figures 4-6 – it would have been useful to show the 'analysed area in context with the broader area studied

AC: We had calibrated the model based on this area, where we had data from all the topography measurement times (2012, March 2013 and June 2013). Thus, the reason for showing the calibration area in these figures is, because we were able to perform the goodness-of-fit only in this area. We have modified the manuscript so that we are not anymore talking about the "analyzed area" but about "calibration area". We have now added the whole simulation area to in the new Figs. 6 and 7, i.e. in the results section related to the morphodynamic analyses. In these two figures, the results of the best simulation are shown. We are now showing the whole simulation area also in Fig. 1. We have moved the former Figs. 4-6 into the supplementary material (there Figs. 1-3), and present a new Fig. 4 (includes parts of former Fig. 6) in the main manuscript.

RC: Page 19 line 4 – ok so how many model runs are now relevant?

AC: We decided to delete that sentence. We present now in the main manuscript only the two best (based on comparison to observed data) model runs (i.e. out of 61 total runs). Altogether 11 runs (out of total 61 runs) which were presented in the original version of the manuscript, have been moved into the supplementary material for showing the results of calibration (see also the earlier responses to all referees).

RC: Page 19 Line 19 – should be satisfactory

AC: During the shortening of the methods section, the sentence had been deleted from the main manuscript.

We have modified this accordingly in the text of supplementary material, where this text had been moved.

RC: Page 19 Lines 6 – 19 – would these plots have been better as hysteresis type plots so plotting Q against volumetric changes in bedload? You have not discussed hysteresis at all? The same comments apply for section 4.2.2 Figure 7 – you need axis labels, especially for the Y axis- what is it showing?

AC: The possible hysteresis has been discussed in the discussion section, and also now mentioned in the results section. As the model was calibrated based on the topographical changes, we wanted to present the results based on the topographical changes. Therefore, we find that plotting Q against bedload would not bring extra value to the manuscript. From the Fig. 5 (earlier figure number 7) it is already possible to see the differences in the timing of the channel change peak and discharge peak. This is already an evidence about the hysteresis. However, we state in the discussion that further studies on this matter are still needed, and that this is just the first evidence of the possibility for the hysteresis effect.

The units of the parameters (Y-axis) had been presented in the legend (former Fig. 7 -> now Fig. 5). We have now added both of the units also in the Y-axis too. We have also added a new Fig. 6 to show the spatial bed elevation changes between the key time steps, which are presented in Fig. 5.

RC: Page 22 – what do you mean reliable? How useful is it for predicting other events/ scenarios?

AC: We have not yet applied the model to larger flood events, as we do not yet have observations of those from Rambla de la Viuda. We will test the model against larger discharges, when this will be possible based on the data. We are waiting a bigger flash

flood to occur in the area. We have also added more discussion about the possibilities of the model to be applied for predicting larger events (throughout the discussion section, e.g. at P21 line31-P22 line 6): "However, despite the high quality data from two events at Rambla de la Viuda, we think that further research with multiple yet-to-come events needs to be run to assess the repeatability and validation of the model even better. For example, at Rambla de la Viuda, large floods have not yet occurred since the beginning of the MLS measurement approaches. As also earlier has been stated (Verhaar et al. 2008; Lotsari et al., 2015), the roughenss conditions defined for small discharge events, might not be suitable for simulating extreme events. Therefore, the work and refinement of the model will continue, and the applicability of the model for larger floods will be tested, when validation data will be available."

RC: Page 23 Lines 1-4 – you definitely need to discuss the applicability of this model to other events a and need to discuss the limitations of this approach!!!

AC: We have modified the discussion section (section number 5) and added there more discussion about the applicability of the model to other larger events/scenarios, and also about the possible limitations. See also our earlier responses.

RC: Page 23 Line 10 – did you really know the hydrograph shape?

AC: We can verify that the hydrograph shape at the study site of Rambla de la Viuda is similar to the hydrograph at Vall d'Alba station. See the supplementary material and also our earlier responses (above) to the similar questions.

RC: Page 25 Lines 17-18 – have you considered how the role of the changing surface structure could be incorporated into the model as this has been shown to have significant impacts in transport rates?

AC: We are not sure if we understood your comment. However, if we understood correctly, the answer is as follows: We had included the longitudinal and transverse slope functions into the model, but otherwise this was out of the scope of the paper of

ours. We did not consider this, as we did not concentrate on coding of the model, but on applying the commercially available version of the model. If looking the model at the study reach scale, we think that the overall transport rate is unrelated to surface structure.

Please also note the supplement to this comment:
https://www.earth-surf-dynam-discuss.net/esurf-2017-52/esurf-2017-52-AC1-supplement.zip

[Figure]

**Fig. 1.**

[Figure]

**Fig. 2.**

**Fig. 3.**

[Figure]

Fig. 4.

[Figure]

[Figure]

**Fig. 5.**

[Figure]

**Fig. 6.**

[Figure]

**Fig. 7.**

---

## Author Response (AR2)

RESPONSE TO REVIEWERS

Thank you for your very helpful and thorough comments. Thank you also for helping with the copy-editing. We have modified the manuscript according to all suggestions made by the reviewer and the Associate Editor. During the thorough proof read, we have copy-edited the manuscript, and also clarified the sentences, wherever needed. We have also added one more funding agency in the acknowledgements section. We also acknowledge the comprehensive reviews done by editors and three reviewers, which helped us to improve the manuscript. The detailed modifications can be seen from the following marked-up manuscript version.

Here are also specific responses to the comments related to the sections, which both reviewer and Associate Editor commented (page numbers are from the marked-up version).

P2 lines 7–9: Both reviewer 1 and Associate Editor commented this sentence. We rephrased the first part of the sentence and also made the second part of the sentence more concise.

P2 line 25: Both reviewer 1 and Associate Editor commented on this sentence. We decided to modify the sentence based on AE's suggestion, i.e. "If the total erosion and deposition is greater…"

P12 line 29: Both reviewer 1 and Associate Editor commented on this sentence. AE suggested to modify it as "was based on", and reviewer 1 had suggested to modify it as "involved adjusting the". We decided to modify the sentence according to reviewer's suggestion.

P15 lines 5–7: Both reviewer 1 and Associate Editor commented that this sentence was unclear. We modified the sentence as suggested, and divided it also into two separate sentences for clarification. The sentences now read as follows: "
[revised manuscript text omitted]